# Message-Passing State-Space Models: Improving Graph Learning with Modern Sequence Modeling

## Abstract

The recent success of State-Space Models (SSMs) in sequence modeling has inspired their adaptation to graph learning. We propose the Message-Passing State-Space Model (MP-SSM), which embeds modern SSM principles directly into the Message-Passing Neural Network (MPNN) framework. This yields a unified methodology for learning on both static and temporal graphs, preserving permutation equivariance and enabling efficient long-range information propagation. Crucially, MP-SSM supports exact sensitivity analysis, allowing us to characterize representational bottlenecks such as vanishing gradients and over-squashing in deep regimes. By combining the representational advantages of SSMs with the structural inductive biases of message passing, MP-SSM contributes to a broader effort of unifying learning principles across architectures. Experiments across synthetic, heterophilic, and spatiotemporal benchmarks demonstrate that our framework produces representations that are both theoretically interpretable and empirically strong. In this sense, MP-SSM provides new insights into the conditions under which distinct neural models converge toward similar representations, advancing the theme of representational unification.

## 1 Introduction

Graph Neural Networks (GNNs), especially Message-Passing Neural Networks (MPNNs), have become a staple in learning from graph-structured data. However, traditional MPNNs like GCNs [61] face challenges in propagating information across distant nodes due to issues such as over-squashing [2, 106, 27] and vanishing gradients [27, 84, 3], which hinder performance in tasks requiring long-range dependency modeling [32]. While various strategies, such as rewiring [106, 60, 49], transformers [64, 119, 88, 33, 31], and weight-space regularization [43, 44], have been proposed to improve signal propagation, a principled and simple solution remains elusive, since most aforementioned methods require substantial architectural modifications and cannot be seamlessly applied to traditional MPNNs like GCN [61]. In parallel, recent breakthroughs in sequence modeling using State-Space Models (SSMs), e.g., LRU [81], S4 [46], and extensions [98, 48, 87, 38], have led to advanced architectures like Mamba [45], Griffin [25], and xLSTM [9]. These models consist of stacked recurrent seq2seq blocks, moving nonlinearities outside the recurrence [4] and interleaving them with multilayer perceptrons (MLPs), enabling long-range dependency modeling, stable gradient flow, efficient training, and universal approximation [80, 77]. This design balances short-term memory retention [59] and nonlinear expressivity [58, 24], a trade-off critical to learn long-term dependencies while representing complex nonlinear relationships within data [85, 110]. Inspired by these advances, researchers have begun adapting SSMs for graph learning. Some approaches adopt spectral methods [57], while others transform graphs into sequences for SSM processing [104, 111, 10], often compromising permutation-equivariance [14] or graph topology. Alternatives like GrassNet [122] rely on spectral decompositions with non-unique modes [69], limiting generality.

Submitted to 39th Conference on Neural Information Processing Systems (NeurIPS 2025). Do not distribute.

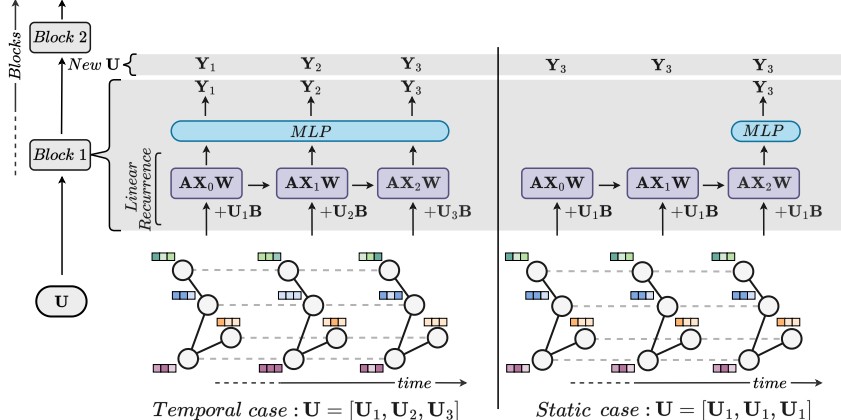

Figure 1: Illustration of our MP-SSM for temporal and static cases, considering a recurrence time $k + 1 = 3$. The temporal case (left) incorporates dynamic updates to node embeddings over time steps, represented as $\mathbf{U} = [\mathbf{U}_1, \mathbf{U}_2, \mathbf{U}_3]$, while the static case (right) uses fixed node embeddings $\mathbf{U} = [\mathbf{U}_1, \mathbf{U}_1, \mathbf{U}_1]$. An MP-SSM block comprises a linear recurrence followed by a multilayer perceptron (MLP). Multiple MP-SSM blocks are stacked to construct a deep MP-SSM architecture.

A comprehensive review of related work is provided in Appendix A. We propose a novel approach that unifies the representational strengths of MPNNs and SSMs by embedding modern state-space heuristics directly into message passing, yielding a principled framework for both static and temporal graphs and providing new insights into when distinct neural architectures converge toward similar internal representations.

**Contributions.** Our work introduces the Message-Passing State-Space Model (MP-SSM):

1. **Unified framework:** Integrates SSMs into MPNNs, preserving permutation equivariance and enabling efficient long-range propagation on both static and temporal graphs.

2. **Theoretical guarantees:** Supports exact Jacobian-based sensitivity analysis, offering precise insights into vanishing gradients and over-squashing.

3. **Empirical performance:** Achieves state-of-the-art results across synthetic, heterophilic, and spatio-temporal benchmarks, with runtime comparable to GCNs.

## 2 Message-Passing State-Space Model

We propose the *Message-Passing State-Space Model* (MP-SSM), which embeds modern SSM principles into message passing. An MP-SSM block consists of a linear state-space recurrence on graphs followed by a graph-agnostic MLP, enabling efficient long-range propagation and parallelization. Due to its popularity and simplicity, we use the symmetrically normalized adjacency with self-loops [61] as graph shift operator (GSO), for our analysis. However, our framework seamlessly extends to any GSO.

Central to our contribution is a linear recurrence over the GSO followed by a shared MLP layer as readout layer. Precisely, we define a block of MP-SSM as a seq2seq model mapping input features $\mathbf{U}_t \in \mathbb{R}^{n \times c'}$ into output states $\mathbf{Y}_t \in \mathbb{R}^{n \times c}$ as

$$\mathbf{X}_{t+1} = \mathbf{A}\mathbf{X}_t\mathbf{W} + \mathbf{U}_{t+1}\mathbf{B}, \qquad t = 0, \dots, k, \tag{1}$$

$$\mathbf{Y}_{t+1} = \text{MLP}(\mathbf{X}_{t+1}), \tag{2}$$

where $\mathbf{X}_t \in \mathbb{R}^{n \times c}$ are the hidden states, $\mathbf{W}, \mathbf{B}$ are learnable weight matrices, and $k$ is an hyperparameter defining the depth of the recurrence. This purely linear recurrence enables exact sensitivity analysis and closed-form parallel implementation. In Appendix E, we describe our fast implementation, discussing both its advantages and limitations, and provide a runtime comparison with a standard GCN, showing that MP-SSM can achieve up to a 1000× speedup. For temporal graphs, $\mathbf{U} = [\mathbf{U}_1, \dots, \mathbf{U}_{k+1}]$; for static graphs, $\mathbf{U} = [\mathbf{U}_1, \dots, \mathbf{U}_1]$, ensuring a unified treatment, see Figure

66 1. Nonlinearity appears only in the MLP, simplifying analysis and computation. In Appendix F
67 we discuss the originality of our method in relation to recent temporal graph that use a state-space
68 modeling approach, like GGRNN [90] and GraphSSM [66].
69 We stack more MP-SSM blocks to develop a hierarchy of representations. Stacking $s$ blocks of
70 depth $k$ yields an effective receptive field of $sk$ hops, supporting stable long-range aggregation. In
71 Appendix G, we provide a multi-hop interpretation of a deep MP-SSM architecture, in the static case.
72 Note that, due to our GSO choice, MP-SSM reduces to a residual GCN when $k = 1$ , see Appendix
73 B, but generalizes beyond it for $k \geq 2$. Standard deep learning heuristics (residuals, normalization,
74 dropout) can be applied between blocks, following modern SSM design. Appendix H presents an
75 ablation study tracing the incremental impact of each SSM heuristic on graph representation learning,
76 progressing from a plain GCN to a deep MP-SSM architecture. Finally, we discuss the complexity
77 and runtimes of MP-SSM in Appendix I.

## 3 Sensitivity Analysis

79 A key advantage of MP-SSM is that its purely linear recurrence allows an *exact* characterization of
80 gradient flow via Jacobians. For node $j$ at step $s$ and node $i$ at step $t \geq s$, the Jacobian of the linear
81 recurrent equation of an MP-SSM block is exactly the following:

$$\frac{\partial \mathbf{X}_t^{(i)}}{\partial \mathbf{X}_s^{(j)}} = \underbrace{(\mathbf{A}^{t-s})_{ij}}_{\text{scalar}} \underbrace{(\mathbf{W}^\top)^{t-s}}_{\text{matrix}}. \tag{3}$$

82 This closed form enables precise analysis of stability and information transfer, allowing us to reason
83 around key challenges in graph learning like over-squashing and vanishing gradients, see Appendix C
84 for a full theoretical analysis. In particular, we can compute a lower bound for the spectral norm of
85 the Jacobian of (3) as follows:

$$\frac{2}{|V| + 2|E|}||\mathbf{W}^{t-s}|| \leq \min_{i,j} \left\| \frac{\partial \mathbf{X}_t^{(i)}}{\partial \mathbf{X}_s^{(j)}} \right\|, \tag{4}$$

86 where $|V|$ and $|E|$ denotes number of vertices and edges, respectively.

87 Regarding over-squashing, we find a class of graph topologies that realise the lower bound in (4), thus
88 representing the worst-case scenario for transferring information. Regarding vanishing gradients, we
89 estimate that a $k$-layer GCN vanishes $2^{-k/2}$ faster than an MP-SSM block of depth $k$. For detailed
90 statements of the theorems, assumptions, and proofs, see Appendix C.

91 Overall, MP-SSM provides a principled theoretical foundation, exact Jacobian computation, provable
92 stability, and precise reasoning about over-squashing and vanishing gradients.

## 4 Experiments

94 We evaluate MP-SSM on static graphs (synthetic shortest-path tasks, Section 4.1 and Appendix K),
95 temporal graphs (spatio-temporal forecasting, Section 4.2 and Appendix L), as well as heterophilic
96 (Appendix N) and long-range real-world benchmarks (Appendix M).

### 4.1 Graph Property Prediction

98 We evaluate MP-SSM on three synthetic
99 tasks from [43], graph diameter, SSSP,
100 and node eccentricity, requiring long-
101 range information flow. Using the orig-
102 inal setup and hyperparameters, Table 1
103 shows MP-SSM outperforms all base-
104 lines, gaining 2.4 points on average, sur-
105 passing A-DGN by 3.4 points on eccen-
106 tricity and exceeding its GCN backbone
107 by over 4 points, demonstrating superior
108 long-range propagation.

Table 1: Mean $log_{10}(\mathrm{MSE})(\downarrow)$ and std averaged on 4 random weight initializations. **First**, **second**, and **third** best results for each task are color-coded.

| Model | Diameter | SSSP | Eccentricity |
|---|---|---|---|
| **MPNNs** | | | |
| A-DGN | **-0.5188**$_{\pm 0.1812}$ | -3.2417$_{\pm 0.0751}$ | **0.4296**$_{\pm 0.1003}$ |
| GAT | 0.8221$_{\pm 0.0752}$ | 0.6951$_{\pm 0.1499}$ | 0.7909$_{\pm 0.0222}$ |
| GCN | 0.7424$_{\pm 0.0466}$ | 0.9499$_{\pm 0.0001}$ | 0.8468$_{\pm 0.0028}$ |
| **Transformers** | | | |
| GPS | **-0.5121**$_{\pm 0.0426}$ | **-3.5990**$_{\pm 0.1949}$ | **0.6077**$_{\pm 0.0282}$ |
| **Ours** | | | |
| MP-SSM | **-3.2353**$_{\pm 0.1735}$ | **-4.6321**$_{\pm 0.0779}$ | **-2.9724**$_{\pm 0.0271}$ |

Table 2: Multivariate time series forecasting on the Metr-LA and PeMS-Bay datasets for Horizon 12. **First**, **second**, and **third** best results for each task are color-coded. Baseline results are reported from [94, 70, 39, 36, 121].

| Model | Metr-LA | | | PeMS-Bay | | |
|---|---|---|---|---|---|---|
| | MAE ↓ | RMSE ↓ | MAPE ↓ | MAE ↓ | RMSE ↓ | MAPE ↓ |
| **Graph Agnostic** | | | | | | |
| HA | 6.99 | 13.89 | 17.54% | 3.31 | 7.54 | 7.65% |
| FC-LSTM | 4.37 | 8.69 | 14.00% | 2.37 | 4.96 | 5.70% |
| SVR | 6.72 | 13.76 | 16.70% | 3.28 | 7.08 | 8.00% |
| VAR | 6.52 | 10.11 | 15.80% | 2.93 | 5.44 | 6.50% |
| **Temporal GNNs** | | | | | | |
| AdpSTGCN | 3.40 | 7.21 | **9.45%** | 1.92 | 4.49 | 4.62% |
| ASTGCN | 6.51 | 12.52 | 11.64% | 2.61 | 5.42 | 6.00% |
| DCRNN | 3.60 | 7.60 | 10.50% | 2.07 | 4.74 | 4.90% |
| GMAN | 3.44 | 7.35 | 10.07% | 1.86 | **4.32** | 4.37% |
| Graph WaveNet | 3.53 | 7.37 | 10.01% | 1.95 | 4.52 | 4.63% |
| GTS | 3.46 | 7.31 | 9.98% | 1.95 | 4.43 | 4.58% |
| MTGNN | 3.49 | 7.23 | 9.87% | 1.94 | 4.49 | 4.53% |
| RGDAN | **3.26** | **7.02** | 9.73% | 1.82 | **4.20** | 4.28% |
| STAEformer | **3.34** | **7.02** | 9.70% | 1.88 | 4.34 | 4.41% |
| STD-MAE | 3.40 | 7.07 | **9.59%** | **1.77** | **4.20** | **4.17%** |
| STEP | 3.37 | **6.99** | 9.61% | **1.79** | **4.20** | **4.18%** |
| STGCN | 4.59 | 9.40 | 12.70% | 2.49 | 5.69 | 5.79% |
| STSGCN | 5.06 | 11.66 | 12.91% | 2.26 | 5.21 | 5.40% |
| **Temporal Graph SSMs** | | | | | | |
| GGRNN | 3.88 | 8.14 | 10.59% | 2.34 | 5.14 | 5.21% |
| GraphSSM-S4 | 3.74 | 7.90 | 10.37% | 1.98 | 4.45 | 4.77% |
| **Ours** | | | | | | |
| MP-SSM | **3.17** | **6.86** | **9.21%** | **1.62** | **4.22** | **4.05%** |

## 4.2 Spatio-Temporal Forecasting

We report here a thorough evaluation of MP-SSM on two popular forecasting datasets, Metr-LA and PeMS-Bay [68], and additional results are provided in Appendix L further three spatio-temporal forecasting benchmarks, namely Chickenpox Hungary, PedalMe London, and Wikipedia math [89]. The aim is to predict future node values from time-series data using original dataset settings. Across both datasets, MP-SSM outperforms existing temporal GNNs, including state-space models GGRNN [90] and GraphSSM [66], highlighting its effectiveness in modeling spatial-temporal dependencies and versatility across static and temporal graph domains.

Full details of the hyperparameter settings for all experiments are described in Appendix O.3. We emphasize that, unlike most state-of-the-art graph models, MP-SSM runs at a speed comparable to that of a standard GCN (see runtime and complexity analyses in Appendix I), even without leveraging the optimized implementation discussed in Appendix E.

## 5 Conclusions

We introduced the Message-Passing State-Space Model (MP-SSM), a framework that unifies modern state-space sequence modeling with message passing on graphs. By embedding SSM principles into MPNNs, MP-SSM achieves efficient and stable information propagation, supports exact sensitivity analysis, and applies broadly across static and temporal domains. Beyond performance gains, our work highlights the representational commonalities between sequence and graph models, illustrating how both families capture dependencies through analogous mechanisms of recurrence and aggregation, despite operating on different data domains. This connection aligns with the broader goal of understanding and unifying neural representations across domains, offering insights into how principles from sequence models can inform graph learning and vice versa.

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

## A  Related Works

**Learning Long-Range Dependencies on Graphs.**  While GNNs effectively model local structures via message passing, they struggle with long-range dependencies due to over-squashing and vanishing gradients [2, 27]. Standard models like GCN [61], GraphSAGE [50], and GIN [118] suffer from degraded performance on tasks requiring global context [5, 32], especially in heterophilic graphs [72, 112]. Solutions include graph rewiring [106, 60], weight-space regularization [43, 44], and physics-inspired dynamics [54]. Graph Transformers (GTs) like SAN [64], Graphormer [119], and GPS [88] enhance expressivity using structural encodings [33, 31], but suffer from quadratic complexity. Scalable alternatives include sparse and linearized attention mechanisms [120, 21, 97, 96, 115, 26], though simple MPNNs often remain competitive [105].

**Learning Spatio-Temporal Interactions on Graphs.** Temporal GNNs often combine GNNs with RNNs to model spatio-temporal dynamics [42]. Some adopt stacked architectures that separate spatial and temporal processing [92, 83, 82, 6, 22], while others integrate GNNs within RNNs for joint modeling [65, 17, 68, 23, 90]. Our approach follows the latter, but goes further by embedding modern SSM principles directly into the GNN architecture, unifying spatial and temporal reasoning through linear recurrence. This contrasts with GGRNN [90], which employs a more elaborate message-passing scheme involving nonlinear aggregation over multiple powers of the graph shift operator at each recurrent step.

**Casting State-Space Models into Graph Learning.**  Several recent models adopt SSMs for static graphs by imposing sequential orderings, e.g., via degree-based sorting [111] or random walks [10], often sacrificing permutation-equivariance. Spectral methods [57] offer alternatives but are computationally demanding and prone to over-squashing [27]. In the temporal graph setting, GraphSSM [66] applies the diffusive dynamics of a GNN backbone first, followed by an SSM as a post-processing module. In contrast, our approach embeds the core principles of modern SSMs directly into the graph learning process, yielding a unified framework that seamlessly supports both static and temporal graph modeling—while maintaining permutation equivariance, computational efficiency, and supporting parallel implementation.

## B  MP-SSM generalizes MPNNs.

We note that our MP-SSM can implement its backbone MPNN, an important property that allows it to retain desired or known behavior from existing MPNNs while also generalizing it and allowing for improved information transfer, as discussed in Section 3. To show that our model can implement its backbone MPNN, which in our case is based on GCN via the chosen GSO, we consider the static case, i.e., an input sequence $[\mathbf{U}_1, \ldots, \mathbf{U}_1]$, under the assumption that the MLP is a nonlinear activation $\sigma$ function. We note that this can be obtained if the weights within the MLP decoder are the identity matrices, i.e., $\mathrm{MLP}(\cdot) = \sigma(\cdot)$. Then an MP-SSM block with $k = 1$ yields a GCN layer. In fact, if $k = 1$ then Equations (1) and (2) read:

$$\mathbf{X}_1 = \mathbf{U}_1\mathbf{B} \quad \Rightarrow \quad \mathbf{X}_2 = \mathbf{A}\mathbf{U}_1\mathbf{B}\mathbf{W} + \mathbf{U}_1\mathbf{B} = \mathbf{A}\mathbf{X}_1\mathbf{W} + \mathbf{X}_1 \quad \Rightarrow \quad \mathbf{Y}_2 = \sigma(\mathbf{A}\mathbf{X}_1\mathbf{W} + \mathbf{X}_1),$$

which implements a GCN with a residual connection. Then $\mathbf{Y}_2$ is passed as an input to the next MP-SSM block, which yields a similar update rule, effectively constructing a deep GCN. However, we note that if $k \geq 2$, then an MP-SSM block deviates from the standard GCN processing.

## C  Detailed Sensitivity Analysis

We conduct a sensitivity analysis of MP-SSM via the spectral norm of the Jacobian of node features, as in [106]. We provide an exact characterization of MP-SSM's gradient flow through the graph, identify unfavourable topological structures that intensify oversquashing effects, and quantitatively assess the impact of removing nonlinearities at each recurrent step of graph diffusion, particularly in alleviating vanishing gradients in the deep regime.

*Remark* C.1. If the GSO is the identity matrix $(\mathbf{A} = \mathbf{I})$, then stacking $s$ MP-SSM blocks with one recurrence each $(k = 1)$ results in a deep MLP of depth $2s$. This feedforward architecture is graph-agnostic, and it can be made resilient to vanishing and exploding gradient issues through standard deep learning heuristics such as residual connections [52] and normalization layers [108], with

dropout being employed as a regularization technique to support the learning of robust hierarchical representations [101]. In our deep MP-SSM architecture, we apply these heuristics between MP-SSM blocks, following established practices in SSMs [46, 45]. Thus, MP-SSM extends graph-agnostic deep feedforward networks, for which established deep learning heuristics are known to effectively address vanishing/exploding gradient issues. This observation motivates our focus for sensitivity analysis on the linear recurrent equation within an MP-SSM block, as it encapsulates the core dynamics relevant to information propagation on graphs. Notably, all the other operations within a deep MP-SSM are independent of the graph structure. Thus, **if the linear recurrent equation supports effective information transfer, then this property naturally extends across the full MP-SSM architecture**, which is fundamentally a stack of such linear recurrences.

Let $\mathbf{X}_s^{(j)}$ and $\mathbf{X}_t^{(i)}$ denote the embeddings of nodes $j$ and $i$ at time steps $s \leq t$. We define:

**Definition C.2** (Local sensitivity). The *local sensitivity* of the features of the $i$-th node to features of the $j$-th node, after $t - s$ applications of message-passing aggregations, is defined as the following spectral norm:

$$\mathcal{S}_{ij}(t - s) = \left\|\left\| \frac{\partial \mathbf{X}_t^{(i)}}{\partial \mathbf{X}_s^{(j)}} \right\|\right\|. \tag{5}$$

Equation (5) measures the influence of node $j$'s features at time $s$ on node $i$ at time $t$.

*Remark* C.3. If the local sensitivity between two nodes increases exponentially with $t - s$, then the learning dynamics of the MPNN are unstable; that is the typical case for linear MPNNs using the adjacency matrix without any normalization or feature normalization. Therefore, **upper bounds on local sensitivity are linked with stable message propagation, in the deep regime**.

The linearity of the recurrence of an MP-SSM block allows an exact computation of the Jacobian between two nodes $j, i$ at different times $s, t$, in terms of the powers of the GSO, as expressed by Equation (6) in Theorem C.4 (for the proof, see Appendix D.2).

**Theorem C.4** (Exact Jacobian computation in MP-SSM). *The Jacobian of the linear recurrent equation of an MP-SSM block, from node $j$ at layer $s$ to node $i$ at layer $t \geq s$, can be computed exactly, and it has the following form:*

$$\frac{\partial \mathbf{X}_t^{(i)}}{\partial \mathbf{X}_s^{(j)}} = \underbrace{(\mathbf{A}^{t-s})_{ij}}_{scalar} \underbrace{(\mathbf{W}^{\top})^{t-s}}_{matrix}. \tag{6}$$

Consequently, GSOs that yield a bounded outcome under iterative multiplication promote stable MP-SSM dynamics, as highlighted in Remark C.3. In Lemma C.5, we formally prove (see Appendix D.1) that the symmetrically normalized adjacency with self-loops exhibits this stability property, along with additional characteristics[1] that support our theoretical analysis.

**Lemma C.5** (Powers of symmetrically normalized adjacency with self-loops). *Assume an undirected graph. The spectrum of the powers of the symmetric normalized adjacency matrix $\mathbf{A} = \mathbf{D}^{-\frac{1}{2}}(\tilde{\mathbf{A}} + \mathbf{I})\mathbf{D}^{-\frac{1}{2}}$ is contained in the interval $[-1, 1]$. The largest eigenvalue of $\mathbf{A}^t$ has absolute value of $1$ with corresponding eigenvector $\mathbf{d} = diag(\mathbf{D}^{\frac{1}{2}})$, for all $t \geq 1$. In particular, the sequence of powers $[\mathbf{A}^t]_{t \geq 1}$ does not diverge or converge to the null matrix.*

Thus, Lemma C.5 implies that the symmetrically normalized adjacency with self-loops serves as a GSO that ensures stable dynamics when performing a large number of message-passing operations in the MP-SSM's framework. Moreover, for such a particular GSO, we can derive a precise approximation of the local sensitivity in the deep regime, as stated in Theorem C.6 and proved in Appendix D.3.

**Theorem C.6** (Approximation deep regime). *Assume a connected graph, and the symmetrically normalized adjacency with self-loops as GSO. Then, for large values of $t - s$, the Jacobian of the linear recurrent equation of an MP-SSM block, from node $j$ at layer $s$ to node $i$ at layer $t \geq s$, admits the following approximation:*

$$\frac{\partial \mathbf{X}_t^{(i)}}{\partial \mathbf{X}_s^{(j)}} \approx \frac{\sqrt{(1 + d_i)(1 + d_j)}}{|V| + 2|E|} (\mathbf{W}^{\top})^{t-s}, \tag{7}$$

---

[1]Similar characteristics of the symmetrically normalized adjacency with self-loops have also been discussed in [79].

where $d_l = \sum_{j=1}^{n} (\tilde{\mathbf{A}})_{lj}$ is the degree of the $l$-th node.

For the case of the symmetrically normalized adjacency with self-loops as GSO, we can find a precise lower bound for the minimum local sensitivity among all possible pairs of nodes in the graph, in the deep regime (proof in Appendix D.4).

**Corollary C.7** (Lower bound minimum sensitivity). *Assume a connected graph, and the symmetrically normalized adjacency with self-loops as GSO. Then, for large values of $t - s$, the following lower bound for the minimum local sensitivity of the linear recurrent equation of an MP-SSM block holds:*

$$\frac{2}{|V| + 2|E|} ||\mathbf{W}^{t-s}|| \leq \min_{i,j} \mathcal{S}_{ij}(t - s). \tag{8}$$

The minimum local sensitivity is realized for pairs of nodes among which the transfer of information is the most critical due to the structure of the graph. Therefore, **lower bounds on the minimum local sensitivity are linked to the alleviation of over-squashing**. Rewiring techniques are known to help combating this phenomenon [27]. Corollary C.7 proves that, without rewiring, MP-SSM can deal with over-squashing by increasing the norm of the recurrent weight matrix. In Remark C.8, we construct an example of a topology that approaches the lower bound of Equation (8), thus realising a worst case scenario due to over-squashing.

*Remark* C.8 (Bottleneck Topologies). A chain of $m$ cliques of order $d$ represents a topology realising a bad scenario for Equation (7), since local sensitivity can reach values as low as $\dfrac{1}{md^2}$, scaling on long chains and large cliques, see Appendix D.3.1 for details. This effect is intrinsically tied to the specific topology of the graph, and it aligns with prior studies that emphasize the challenges of learning on graphs with bottleneck structures [106].

To assess the overall gradient information flow across the entire graph in the deep regime, we define:

**Definition C.9** (Global sensitivity). The *global sensitivity* of node features of the overall graph after $t - s$ hops of message aggregation is defined as:

$$\mathcal{S}(t - s) = \max_{i,j} \mathcal{S}_{ij}(t - s). \tag{9}$$

*Remark* C.10. The local sensitivity between two far-apart nodes can be physiologically small due to the particular topology of the graph (e.g. bottlenecks), or it can be even 0 if two nodes are not connected by any walk. However, if the local sensitivity converges to 0, in the deep regime of large $t - s$, for all the pairs of nodes, i.e., if the global sensitivity converges to 0 regardless of the particular topology of the graph, then it means that the MPNN model is characterized by a vanishing information flow. Therefore, **lower bounds on global sensitivity are linked to the alleviation of vanishing gradient issues, in the deep regime**.

For connected graphs, we can leverage the exact Jacobian computation of Theorem C.4 to prove the following lower bound on the global sensitivity, see Appendix D.5 for the proof.

**Theorem C.11** (Lower bound global sensitivity). *Assume a connected graph. The global sensitivity of the linear recurrent equation of an MP-SSM block is lower bounded as follows:*

$$\frac{\rho(\mathbf{A})^{t-s}}{|V|} ||\mathbf{W}^{t-s}|| \leq \mathcal{S}(t - s), \tag{10}$$

*where $\rho(\mathbf{A})$ is the spectral radius of the GSO. Thus, for the symmetrically normalized adjacency with self-loops, it holds the lower bound $\dfrac{1}{|V|} ||\mathbf{W}^{t-s}|| \leq \mathcal{S}(t - s)$.*

This theoretical result demonstrates that MP-SSM ensures values of the global sensitivity strictly greater than zero, for any depth $t - s$ and for connected graphs with any number of nodes. This result cannot be guaranteed in a standard MPNN, as the nonlinearity applied at each time step increasingly contributes to vanish information as the depth increases. We provide an extended discussion about this point in Appendix J.

*Remark* C.12. Note that both results of Equation (6) and Equation (10) hold for any GSO. However, for the particular case of the symmetrically normalized adjacency with self-loops, we can provide more precise approximations and bounds.

From Section B, we know that MP-SSM generalizes its backbone MPNNs, and the GCN architecture in particular when using the symmetrically normalized adjacency with self-loops as GSO. In Theorem C.13, we provide an estimation of the vanishing effect caused by the application at each time step of a ReLU nonlinearity in a standard GCN compared with our MP-SSM, in the deep regime, as we prove in Appendix D.6.

**Theorem C.13** (GCN vanishes more than MP-SSM). *Let us consider a GCN network that aggregates information from $k$ hops away, i.e., with $k$ layers, equipped with the ReLU activation function. Then, the GCN vanishes information at a $2^{-\frac{k}{2}}$ faster rate than our MP-SSM block with $k$ linear recurrent steps.*

# D Proofs

Here, we provide all the proofs of lemmas, theorems, and corollaries stated in the main text.

## D.1 Proof of Lemma C.5

**Lemma.** Assume an undirected graph. The spectrum of the powers of the symmetric normalized adjacency matrix $\mathbf{A} = \mathbf{D}^{-\frac{1}{2}}(\tilde{\mathbf{A}} + \mathbf{I})\mathbf{D}^{-\frac{1}{2}}$ is contained in the interval $[-1, 1]$. The largest eigenvalue of $\mathbf{A}^t$ has absolute value of 1 with corresponding eigenvector $\mathbf{d} = \mathrm{diag}(\mathbf{D}^{\frac{1}{2}})$, for all $t \geq 1$. In particular, the sequence of powers $[\mathbf{A}^t]_{t \geq 1}$ does not diverge or converge to the null matrix.

*Proof.* $\mathbf{A}^t = \left(\mathbf{D}^{-\frac{1}{2}}(\tilde{\mathbf{A}} + \mathbf{I})\mathbf{D}^{-\frac{1}{2}}\right)\left(\mathbf{D}^{-\frac{1}{2}}(\tilde{\mathbf{A}} + \mathbf{I})\mathbf{D}^{-\frac{1}{2}}\right)\ldots\left(\mathbf{D}^{-\frac{1}{2}}(\tilde{\mathbf{A}} + \mathbf{I})\mathbf{D}^{-\frac{1}{2}}\right) = \mathbf{D}^{-\frac{1}{2}}(\tilde{\mathbf{A}} + \mathbf{I})\left(\mathbf{D}^{-1}(\tilde{\mathbf{A}} + \mathbf{I})\right)^{t-1}\mathbf{D}^{-\frac{1}{2}}$. Now, $\mathbf{D}^{-1}(\tilde{\mathbf{A}} + \mathbf{I})$ is a stochastic matrix, and so also its powers are stochastic matrices. Therefore, $\mathbf{D}^{-\frac{1}{2}}\mathbf{A}^t\mathbf{D}^{\frac{1}{2}} = \left(\mathbf{D}^{-1}(\tilde{\mathbf{A}} + \mathbf{I})\right)^t$ is a stochastic matrix. The eigenvalues of a stochastic matrix are contained in the closed unitary disk [75, 8]. Let, $\lambda_1, \ldots, \lambda_n$ all the eigenvalues (not necessarily distinct) of such a stochastic matrix, with corresponding eigenvectors $\mathbf{v}_1, \ldots, \mathbf{v}_n$. Thus, $\mathbf{D}^{-\frac{1}{2}}\mathbf{A}^t\mathbf{D}^{\frac{1}{2}}\mathbf{v}_l = \lambda_l\mathbf{v}_l$, from which it follows, multiplying both sides by $\mathbf{D}^{\frac{1}{2}}$, that $\mathbf{A}^t\mathbf{D}^{\frac{1}{2}}\mathbf{v}_l = \lambda_l\mathbf{D}^{\frac{1}{2}}\mathbf{v}_l$. This means that the eigenvalues of $\mathbf{A}^t$ are exactly the same of those of the stochastic matrix $\mathbf{D}^{-\frac{1}{2}}\mathbf{A}^t\mathbf{D}^{\frac{1}{2}}$ with eigenvectors $\mathbf{D}^{\frac{1}{2}}\mathbf{v}_1, \ldots, \mathbf{D}^{\frac{1}{2}}\mathbf{v}_n$, for all $t$. In particular, the assumption of undirected graph implies $\mathbf{A}$ is a symmetric matrix, thus we get that all eigenvalues of $\mathbf{A}^t$ are real and contained inside $[-1, 1]$, for all $t$. Since the spectral radius of a stochastic matrix is 1, and the vector $\mathbf{1}$ with all components equal to 1 is necessarily an eigenvector due to the row-sum being 1 for a stochastic matrix, then it follows that the largest eigenvalue of $\mathbf{A}^t$ is 1 and $\mathbf{d} = \mathrm{diag}(\mathbf{D}^{\frac{1}{2}})$ is an eigenvector corresponding to eigenvalue 1, for all $t$.

To see why the sequence of powers $[\mathbf{A}^t]_{t \geq 1}$ does not diverge or converge to the null matrix, we observe that, since $\mathbf{A}$ is symmetric, the Spectral Theorem implies we can diagonalize in $\mathbb{R}$ the matrix $\mathbf{A} = \mathbf{Q}\mathbf{\Lambda}\mathbf{Q}^{\top}$ with $\mathbf{Q}$ orthogonal matrix and $\mathbf{\Lambda}$ diagonal matrix of real eigenvalues. Powers of $\mathbf{A}$ can be written as $\mathbf{A}^t = (\mathbf{Q}\mathbf{\Lambda}\mathbf{Q}^{\top})(\mathbf{Q}\mathbf{\Lambda}\mathbf{Q}^{\top})\ldots(\mathbf{Q}\mathbf{\Lambda}\mathbf{Q}^{\top}) = \mathbf{Q}\mathbf{\Lambda}^t\mathbf{Q}^{\top}$. Thus the eigenvalues of $\mathbf{A}^t$ are $\lambda_l^t$, for $l = 1, \ldots, n$. We already proved that the eigenvalues $\lambda_n \leq \ldots \leq \lambda_1$ are contained in the real interval $[-1, 1]$. Hence, this ensures that the sequence of powers cannot diverge. On the other hand, we can spectrally decompose symmetric matrices as follows [51], $\mathbf{A}^t = \sum_{l=1}^{n} \lambda_l^t\mathbf{q}_l\mathbf{q}_l^{\top}$, where $\mathbf{q}_l$ is the eigenvector corresponding to the eigenvalue $\lambda_l$. Thus, for large values of $t$, the spectral components corresponding to eigenvalues strictly less than 1 in absolute value vanish, so the matrix $\mathbf{A}^t$ approaches the sum of terms corresponding to eigenvalues with absolute value equal to 1. This proves that the sequence of powers cannot converge to the null matrix. $\square$

## D.2 Proof of Theorem C.4

**Theorem.** The Jacobian of the linear recurrent equation of an MP-SSM block, from node $j$ at layer $s$ to node $i$ at layer $t \geq s$, can be computed exactly, and it has the following form:

$$\frac{\partial \mathbf{X}_t^{(i)}}{\partial \mathbf{X}_s^{(j)}} = \underbrace{(\mathbf{A}^{t-s})_{ij}}_{\text{scalar}} \underbrace{(\mathbf{W}^{\top})^{t-s}}_{\text{matrix}}.$$

*Proof.* In this proof we use the notation $(\mathbf{M})_{ij}$ to denote the $(i, j)$ entry of a matrix $\mathbf{M}$, and $\mathbf{M}^{(i)}$ to denote the $i$-th row of a matrix $\mathbf{M}$. Let us start with the recurrent equation $\mathbf{X}_{t+1} = \mathbf{A}\mathbf{X}_t\mathbf{W} + \mathbf{U}_{t+1}\mathbf{B}$. Therefore, the $i$-th node features are updated as follows: $\mathbf{X}_{t+1}^{(i)} = \sum_{l=1}^{n}(\mathbf{A})_{il}\mathbf{X}_t^{(l)}\mathbf{W} + \mathbf{U}_{t+1}^{(i)}\mathbf{B}$.

Now, the only term involving $\mathbf{X}_t^{(j)}$ is $(\mathbf{A})_{ij}\mathbf{X}_t^{(j)}\mathbf{W}$. Therefore, the Jacobian reads $\dfrac{\partial \mathbf{X}_{t+1}^{(i)}}{\partial \mathbf{X}_t^{(j)}} =$

$\dfrac{\partial}{\partial \mathbf{X}_t^{(j)}}\left((\mathbf{A})_{ij}\mathbf{X}_t^{(j)}\mathbf{W}\right)$. Now, given a row vector $\mathbf{x} \in \mathbb{R}^c$ and a square matrix $\mathbf{M}$, then the function

$\mathbf{f}(\mathbf{x}) = \mathbf{x}\mathbf{M}$, whose $i$-th component is $f_i = \sum_{l=1}^{c}x_l(\mathbf{M})_{li}$, has derivatives $\frac{\partial f_i}{\partial \mathbf{x}_j} = \frac{\partial}{\partial x_j}(x_j(\mathbf{M})_{ji}) =$

$(\mathbf{M})_{ji}$. Hence, the Jacobian is $\frac{\partial \mathbf{f}}{\partial \mathbf{x}} = \mathbf{M}^\top$. Therefore, it holds $\dfrac{\partial \mathbf{X}_{t+1}^{(i)}}{\partial \mathbf{X}_t^{(j)}} = (\mathbf{A})_{ji}\mathbf{W}^\top$. For the case of non-consecutive time steps, we can unfold the recurrent equation $\mathbf{X}_{t+1} = \mathbf{A}\mathbf{X}_t\mathbf{W} + \mathbf{U}_{t+1}\mathbf{B}$ between any two time steps $s \leq t$, as follows:

$$\mathbf{X}_t = \mathbf{A}^{t-s}\mathbf{X}_s\mathbf{W}^{t-s} + \sum_{l=0}^{t-s-1}\mathbf{A}^l\mathbf{U}_{t-l}\mathbf{B}\mathbf{W}^i. \tag{11}$$

From the unfolded recurrent equation (11) of a MP-SSM we can see that the only term involving $\mathbf{X}_s$ is $\mathbf{A}^{t-s}\mathbf{X}_s\mathbf{W}^{t-s}$. Thus, the Jacobian reads $\dfrac{\partial \mathbf{X}_t^{(i)}}{\partial \mathbf{X}_s^{(j)}} = \dfrac{\partial}{\partial \mathbf{X}_s^{(j)}}\left((\mathbf{A}^{t-s}\mathbf{X}_s\mathbf{W}^{t-s})^{(i)}\right) =$

$\dfrac{\partial}{\partial \mathbf{X}_s^{(j)}}\left((\mathbf{A}^{t-s})_{ij}\mathbf{X}_s^{(j)}\mathbf{W}^{t-s}\right) = (\mathbf{A}^{t-s})_{ij}(\mathbf{W}^\top)^{t-s}$.

$\square$

## D.3 Proof of Theorem C.6

**Theorem.** Assume a connected graph, and the symmetrically normalized adjacency with self-loops as GSO. Then, for large values of $t - s$, the Jacobian of the linear recurrent equation of an MP-SSM block, from node $j$ at layer $s$ to node $i$ at layer $t \geq s$, admits the following approximation:

$$\frac{\partial \mathbf{X}_t^{(i)}}{\partial \mathbf{X}_s^{(j)}} \approx \frac{\sqrt{(1 + d_i)(1 + d_j)}}{|V| + 2|E|}(\mathbf{W}^\top)^{t-s},$$

where $d_l = \sum_{j=1}^{n}(\tilde{\mathbf{A}})_{lj}$ is the degree of the $l$-th node.

*Proof.* We provide an estimation of the term $(\mathbf{A}^{t-s})_{ij}$ for the case of large values of $t - s$, and assuming a connected graph. We use the decomposition $\mathbf{A}^{t-s} = \sum_{l=1}^{n}\lambda_l^{t-s}\mathbf{q}_l\mathbf{q}_l^\top$, where $\mathbf{q}_l$ is the unitary eigenvector corresponding to the eigenvalue $\lambda_l$. As discussed in the proof of Lemma C.5, for large values of $t - s$, all the spectral components corresponding to eigenvalues strictly less than 1 (in absolute value) tend to converge to 0. Moreover, by the Perron–Frobenius theorem for irreducible non-negative matrices [55], since the graph is connected and with self-loops, there is only one simple eigenvalue equal to 1, and $-1$ cannot be an eigenvalue. Thus it holds the approximation $\mathbf{A}^{t-s} \approx \mathbf{q}_1\mathbf{q}_1^\top$. Now thanks to Lemma C.5, we know that $\mathbf{q}_1$ must be the vector $\mathbf{d} = \text{diag}(\mathbf{D}^{\frac{1}{2}})$ normalised to be unitary, and $\mathbf{D}$ is the degree matrix of $\tilde{\mathbf{A}} + \mathbf{I}$. Thus, $\mathbf{q}_1 = \dfrac{(\sqrt{1 + d_1}, \ldots, \sqrt{1 + d_n})}{\sqrt{\sum_{l=1}^{n}(1 + d_l)}}$, where $d_l = \sum_{j=1}^{n}(\tilde{\mathbf{A}})_{lj}$ is the degree of the $l$-th node. Therefore, $(\mathbf{q}_1\mathbf{q}_1^\top)_{ij} = \dfrac{\sqrt{(1 + d_i)(1 + d_j)}}{n + \sum_{l=1}^{n}d_l} = \dfrac{\sqrt{(1 + d_i)(1 + d_j)}}{|V| + 2|E|}$.

$\square$

### D.3.1 Example of a bad scenario for Equation (7)

Figure 2 illustrates an example of a bad scenario for Equation (7), i.e., a chain of $m$ cliques of order $d$ connected via bridge-nodes of degree 2 (the minimum to connect them). In the Figure, we consider

731 $m = 6$ and $d = 10$. The pair of bridge nodes $i$ and $j$ depicted in red in Figure 2 are 12 hops apart, so
732 it can be considered a relatively long-term interaction.

733 In the long-term approximation given by Equation (7), the local sensitivity between two bridge
734 nodes of this topology scales as $\frac{1}{md^2}$, for long chains ($m$ large) and big cliques ($d$ large). In fact, in
735 such a graph the vast majority of nodes has degree approximately $d - 1$, thus $\sum_{l=1}^{n} d_l \approx n(d - 1)$.
736 Specifically, there are exactly $m - 1$ nodes of degree 2 (bridge nodes), and $md$ nodes with degree
737 approximately $d - 1$. Now, $n = m - 1 + md \approx md$, therefore $n + \sum_{l=1}^{n} d_l \approx n + n(d - 1) =$
738 $nd \approx md^2$. Scaling to long chains and large cliques, this approximation becomes more accurate, and
739 so the local sensitivity between two bridge nodes is rescaled by the term $\frac{\sqrt{(1+d_i)(1+d_j)}}{n+\sum_{l=1}^{n} d_l} \approx \frac{3}{md^2}$.

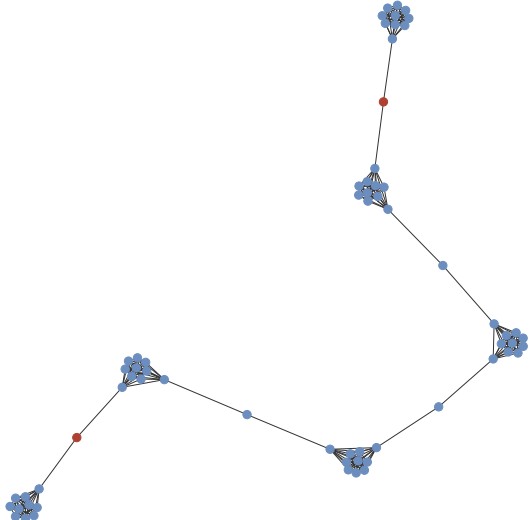

Figure 2: A chain of six cliques (containing ten nodes each) connected via bridge-nodes of degree 2.
The pair of red nodes is a pair of nodes that minimizes the quantity in Equation (7). Note that the red
nodes are 12 hops apart, so it can be considered long-term.

## D.4 Proof of Corollary C.7

741 **Corollary.** Assume a connected graph, and the symmetrically normalized adjacency with self-loops
742 as GSO. Then, for large values of $t - s$, the following lower bound for the minimum local sensitivity
743 of the linear recurrent equation of an MP-SSM block holds:

$$\frac{2}{|V| + 2|E|}||\mathbf{W}^{t-s}|| \leq \min_{i,j} \mathcal{S}_{ij}(t - s). \tag{12}$$

*Proof.* In the deep regime, we can use the approximation of Equation (7) of $\frac{\partial \mathbf{X}_t^{(i)}}{\partial \mathbf{X}_s^{(j)}} \approx$

$\frac{\sqrt{(1 + d_i)(1 + d_j)}}{|V| + 2|E|}(\mathbf{W}^\top)^{t-s}$. Therefore, we have:

$$\min_{i,j} \left\|\frac{\partial \mathbf{X}_t^{(i)}}{\partial \mathbf{X}_s^{(j)}}\right\| \approx \frac{1}{|V| + 2|E|}\left\|(\mathbf{W}^\top)^{t-s}\right\| \min_{i,j} \sqrt{(1 + d_i)(1 + d_j)} \geq \frac{2}{|V| + 2|E|}\left\|(\mathbf{W}^\top)^{t-s}\right\|,$$

744 where the last inequality holds since the minimum degree value of a node in a connected graph
745 is 1. Thus, we conclude that $\min_{i,j} \mathcal{S}_{ij}(t - s) \geq \frac{2}{|V| + 2|E|}||(\mathbf{W}^\top)^{t-s}|| = \frac{2}{|V| + 2|E|}||\mathbf{W}^{t-s}||$,
746 noticing that $||\mathbf{W}^\top|| = ||\mathbf{W}||$. $\qquad\square$

## D.5 Proof of Theorem C.11

**Theorem.** Assume a connected graph. The global sensitivity of the linear recurrent equation of an MP-SSM block is lower bounded as follows:

$$\mathcal{S}(t-s) \geq \frac{\rho(\mathbf{A})^{t-s}}{|V|} ||\mathbf{W}^{t-s}||,$$

where $\rho(\mathbf{A})$ is the spectral radius of the GSO. Thus, for the symmetrically normalized adjacency with self-loops, it holds the lower bound $\mathcal{S}(t-s) \geq \frac{1}{|V|} ||\mathbf{W}^{t-s}||$.

*Proof.* By Equations (5), (6) and (9), we get $\mathcal{S}(t-s) = \max_{i,j} |(\mathbf{A}^{t-s})_{ij}| || (\mathbf{W}^{\top})^{t-s}|| = \max_{i,j} |(\mathbf{A}^{t-s})_{ij}| || \mathbf{W}^{t-s}||$. Let us define $n = |V|$ the number of nodes. The square of the maximum entry of an $(n,n)$ matrix $\mathbf{M}$ is always greater than the arithmetic mean of all the square coefficients, in other words, $\frac{||\mathbf{M}||_F^2}{n^2} \leq \max_{i,j} \mathbf{M}_{i,j}^2$, where $||\mathbf{M}||_F$ denotes the Frobenius norm. Therefore, $\frac{||\mathbf{M}||_F}{n} \leq \max_{i,j} |\mathbf{M}_{i,j}|$. Now, the symmetry of $\mathbf{A}$ implies there are $\lambda_1, \ldots, \lambda_n$ real eigenvalues with corresponding orthonormal eigenvectors $\mathbf{q}_1, \ldots, \mathbf{q}_n$ so that we can decompose $\mathbf{A}^{t-s} = \sum_{l=1}^{n} \lambda_l^{t-s} \mathbf{q}_l \mathbf{q}_l^{\top}$. Thus, the Frobenius norm is $||\mathbf{A}^{t-s}||_F = \sqrt{\sum_{l=1}^{n} \lambda_l^{2(t-s)} ||\mathbf{q}_l||^2} = \sqrt{\sum_{l=1}^{n} \lambda_l^{2(t-s)}} \geq |\lambda_1|^{t-s}$, where $|\lambda_1|$ is the largest in absolute value between all the eigenvalues, i.e. the spectral radius $\rho(\mathbf{A})$.

$$\max_{i,j} |(\mathbf{A}^{t-s})_{ij}| \geq \frac{||\mathbf{A}^{t-s}||_F}{n} \geq \frac{\rho(\mathbf{A})^{t-s}}{n}, \tag{13}$$

from which we get the thesis

$$\mathcal{S}(t-s) = \max_{i,j} |(\mathbf{A}^{t-s})_{ij}| || \mathbf{W}^{t-s}|| \geq \frac{\rho(\mathbf{A})^{t-s}}{n} ||\mathbf{W}^{t-s}||.$$

For the particular case of symmetrically normalized adjacency with self-loops, the spectral radius $\rho(\mathbf{A})$ is exactly 1 due to Lemma C.5. $\square$

## D.6 Proof of Theorem C.13

**Theorem.** Let us consider a GCN network that aggregates information from $k$ hops away, i.e., with $k$ layers, equipped with the ReLU activation function. Then, the GCN vanishes information at a $2^{-\frac{k}{2}}$ faster rate than our MP-SSM block with a number $k$ of linear recurrent steps.

*Proof.* The state-update equation of a GCN with a residual connection is $\mathbf{X}_{t+1} = \sigma(\mathbf{A}\mathbf{X}_t\mathbf{W} + \mathbf{X}_t)$. Therefore, the features of $i$-th node at time $t+1$ are updated as $\mathbf{X}_{t+1}^{(i)} = \sigma\left(\sum_{l=1}^{n} (\mathbf{A})_{il} \mathbf{X}_t^{(l)} \mathbf{W} + \mathbf{X}_t^{(i)}\right)$. Similarly to the proof of theorem C.4, we can write

$$\frac{\partial \mathbf{X}_{t+1}^{(i)}}{\partial \mathbf{X}_t^{(j)}} = \frac{\partial}{\partial \mathbf{X}_t^{(j)}} \left( \sigma\left((\mathbf{A})_{ij} \mathbf{X}_t^{(j)} \mathbf{W}\right) \right) =$$

$$= \text{diag}\left( \sigma'\left((\mathbf{A})_{ij} \mathbf{X}_t^{(j)} \mathbf{W}\right) \right) (\mathbf{A})_{ij} \mathbf{W}^{\top},$$

where we assumed that $i \neq j$, so that the residual connection term does not appear in the derivative w.r.t. $\mathbf{X}_t^{(j)}$. Since we are considering $\sigma = \text{ReLU}$, the diagonal entries $\sigma'\left((\mathbf{A})_{ij} \mathbf{X}_t^{(j)} \mathbf{W}\right)$ are either 0 or 1. Let's assume that the components of the vector $\sigma'\left((\mathbf{A})_{ij} \mathbf{X}_t^{(j)} \mathbf{W}\right)$ are independent and identically distributed (i.i.d.) Bernoulli random variables, each with probability $\frac{1}{2}$ of taking the value 0. Now, let's consider a walk $\{(i_t, j_t)\}_{t=0}^{k-1}$ of length $k$ connecting the $j$-th node at a reference time $t = 0$ to the $i$-th node at time $t = k$. Then, the Jacobian of GCN along such a walk reads:

$$\frac{\partial \mathbf{X}_k^{(i)}}{\partial \mathbf{X}_0^{(j)}} = \prod_{t=0}^{k-1} \mathbf{P}_t \mathbf{M}_t,$$

where $\mathbf{P}_t = \mathrm{diag}\left( \sigma'\left( (\mathbf{A})_{i_t j_t} \mathbf{X}_t^{(j_t)} \mathbf{W} \right) \right)$, and $\mathbf{M}_t = (\mathbf{A})_{i_t j_t} \mathbf{W}^\top$. On the other hand, the Jacobian of the linear recurrent equation (1) of an MP-SSM block, in the static case with a number $k$ of linear recurrent steps computed along the same walk reads:

$$\frac{\partial \mathbf{X}_k^{(i)}}{\partial \mathbf{X}_0^{(j)}} = \prod_{t=0}^{k-1} \mathbf{M}_t.$$

We aim to prove that, for a generic vector $\mathbf{x}$ with entries i.i.d. random variables distributed symmetrically about zero (e.g. according to a Normal distribution with zero mean), it holds the approximation $|| \prod_{t=0}^{k-1} \mathbf{P}_t \mathbf{M}_t \mathbf{x}|| \approx 2^{-\frac{k}{2}} || \prod_{t=0}^{k-1} \mathbf{M}_t \mathbf{x}||$. We prove the thesis using a recursive argument. First, we observe that, denoting $\mathbf{y} = \mathbf{M}_0 \mathbf{x}$, then we can write

$$||\mathbf{P}_0 \mathbf{M}_0 \mathbf{x}||^2 = ||\mathbf{P}_0 \mathbf{y}||^2 = (p_1 y_1)^2 + \ldots + (p_n y_n)^2. \tag{14}$$

Now, since the $p_i$ are assumed i.i.d. Bernoulli random variables, each with probability $\frac{1}{2}$ of taking the value 0, in the sum of (14), roughly a portion of half of the contributions from $\mathbf{y}$ are zeroed-out due to action of $\mathbf{P}_0$. Therefore,

$$||\mathbf{P}_0 \mathbf{M}_0 \mathbf{x}||^2 = ||\mathbf{P}_0 \mathbf{y}||^2 \approx \frac{1}{2}||\mathbf{y}||^2 = \frac{1}{2}||\mathbf{M}_0 \mathbf{x}||^2. \tag{15}$$

Note that the larger the dimension of the graph $n$, the more accurate the approximation of (15). Therefore, we conclude that $||\mathbf{P}_0 \mathbf{M}_0 \mathbf{x}|| \approx 2^{-\frac{1}{2}}||\mathbf{M}_0 \mathbf{x}||$. Now, we proceed recursively by denoting $\tilde{\mathbf{x}}_t = \mathbf{P}_{t-1} \mathbf{M}_{t-1} \ldots \mathbf{P}_0 \mathbf{M}_0 \mathbf{x}$, and defining the scalars $c_t = \dfrac{||\mathbf{M}_t \tilde{\mathbf{x}}_t||}{||\tilde{\mathbf{x}}_t||} > 0$, for all $t = 1, \ldots, k-1$. Then, we can write

$$||\mathbf{P}_{k-1} \mathbf{M}_{k-1} \mathbf{P}_{k-2} \mathbf{M}_{k-2} \ldots \mathbf{P}_0 \mathbf{M}_0 \mathbf{x}|| =$$
$$= ||\mathbf{P}_{k-1} \mathbf{M}_{k-1} \tilde{\mathbf{x}}_{k-1}|| \approx$$
$$\approx 2^{-\frac{1}{2}} ||\mathbf{M}_{k-1} \tilde{\mathbf{x}}_{k-1}|| =$$
$$= 2^{-\frac{1}{2}} c_{k-1} ||\tilde{\mathbf{x}}_{k-1}|| =$$
$$= 2^{-\frac{1}{2}} c_{k-1} ||\mathbf{P}_{k-2} \mathbf{M}_{k-2} \tilde{\mathbf{x}}_{k-2}|| \approx$$
$$\approx 2^{-\frac{1}{2}} c_{k-1} 2^{-\frac{1}{2}} c_{k-2} ||\tilde{\mathbf{x}}_{k-2}|| \approx \ldots$$
$$\approx 2^{-\frac{k}{2}} c_{k-1} c_{k-2} \ldots c_0 ||\mathbf{x}||.$$

On the other hand, for the case of MP-SSM, it reads:

$$||\mathbf{M}_{k-1} \mathbf{M}_{k-2} \ldots \mathbf{M}_0 \mathbf{x}|| = c_{k-1} ||\mathbf{M}_{k-2} \ldots \mathbf{M}_0 \mathbf{x}|| =$$
$$= c_{k-1} c_{k-2} ||\mathbf{M}_{k-3} \ldots \mathbf{M}_0 \mathbf{x}|| = \ldots$$
$$= c_{k-1} c_{k-2} \ldots c_0 ||\mathbf{x}||.$$

This proves that a standard GCN vanishes information $2^{-\frac{k}{2}}$ faster than MP-SSM. We assumed weight sharing in the GCN, but the same proof holds assuming different weights $\mathbf{W}_1, \ldots, \mathbf{W}_k$ at each GCN layer, by simply using the same exact weight matrices for the linear equation of MP-SSM. $\qquad\square$

# E Fast Parallel Implementation

We describe all the details to derive and implement a fast parallel implementation for the computation of an MP-SSM block.

The unfolded recurrence of an MP-SSM block gives the following closed-form solution:

$$\mathbf{X}_{k+1} = \mathbf{A}^k \mathbf{U}_1 \mathbf{B} \mathbf{W}^k + \mathbf{A}^{k-1} \mathbf{U}_2 \mathbf{B} \mathbf{W}^{k-1} + \ldots + \mathbf{A} \mathbf{U}_k \mathbf{B} \mathbf{W} + \mathbf{U}_{k+1} \mathbf{B}. \tag{16}$$

Therefore the equation of an MP-SSM block reads:

$$\mathbf{X}_{k+1} = \sum_{i=0}^{k} \mathbf{A}^i \mathbf{U}_{k+1-i} \mathbf{B} \mathbf{W}^i, \tag{17}$$

$$\mathbf{Y}_{k+1} = \mathrm{MLP}(\mathbf{X}_{k+1}), \tag{18}$$

The closed-form solution of an MP-SSM block tells us that we could implement the whole recurrence in one shot. However, the computation of the powers of both the GSO, $\mathbf{A}$, and the recurrent weights, $\mathbf{W}$, can be extremely expensive for generic matrices and large values of $k$. On the other hand, the powers of diagonal matrices are fairly easy to compute, since they are simply the powers of their diagonal entries. Below, we show how to reduce a generic dense real-valued MP-SSM block to an equivalent diagonalised complex-valued MP-SSM block.

Assume the following diagonalisation of the shift operator: $\mathbf{A} = \mathbf{P}\boldsymbol{\Lambda}\mathbf{P}^{-1}$. If undirected graph, i.e., $\mathbf{A}$ is symmetric, then by spectral theorem the $\mathbf{P}$ is a real orthogonal matrix (i.e. $\mathbf{P}^{-1} = \mathbf{P}^{\top}$) and $\boldsymbol{\Lambda}$ is real.

Assume the following diagonalisation of the weights: $\mathbf{W} = \mathbf{V}\boldsymbol{\Sigma}\mathbf{V}^{-1}$. If using dense real matrices as weights, then their diagonalisation is possible only assuming complex matrices of eigenvectors $\mathbf{V}$ and complex eigenvalues $\boldsymbol{\Sigma}$. Also, note that the set of defective matrices (i.e. non-diagonalizable in $\mathbb{C}$) has zero Lebesgue measure [41].

Assume the following MLP equations with 2 layers: $\mathrm{MLP}(\mathbf{X}) = \phi(\mathbf{X}\mathbf{W}_1)\mathbf{W}_2$, where $\phi$ is a nonlinearity, and $\mathbf{W}_1, \mathbf{W}_2$ real dense matrices.

With the above assumptions, the MP-SSM block equations can be equivalently written as:

$$\mathbf{X}_{k+1} = \sum_{i=0}^{k} \mathbf{P}\boldsymbol{\Lambda}^i \mathbf{P}^{-1}\mathbf{U}_{k+1-i}\mathbf{B}\mathbf{V}\boldsymbol{\Sigma}^i\mathbf{V}^{-1}, \tag{19}$$

$$\mathbf{Y}_{k+1} = \phi(\mathbf{X}_{k+1}\mathbf{W}_1)\mathbf{W}_2, \tag{20}$$

which we can write as:

$$\mathbf{X}_{k+1} = \mathbf{P}\left(\sum_{i=0}^{k} \boldsymbol{\Lambda}^i \mathbf{P}^{-1}\mathbf{U}_{k+1-i}\mathbf{B}\mathbf{V}\boldsymbol{\Sigma}^i\right)\mathbf{V}^{-1}, \tag{21}$$

$$\mathbf{Y}_{k+1} = \phi(\mathbf{X}_{k+1}\mathbf{W}_1)\mathbf{W}_2, \tag{22}$$

Multiply on the left side both terms by $\mathbf{P}^{-1}$ and on the right side both terms by $\mathbf{V}$

$$\mathbf{P}^{-1}\mathbf{X}_{k+1}\mathbf{V} = \sum_{i=0}^{k} \boldsymbol{\Lambda}^i \mathbf{P}^{-1}\mathbf{U}_{k+1-i}\mathbf{B}\mathbf{V}\boldsymbol{\Sigma}^i \tag{23}$$

If we change coordinate reference to $\mathbf{Z}_{k+1} = \mathbf{P}^{-1}\mathbf{X}_{k+1}\mathbf{V}$, then we can write:

$$\mathbf{Z}_{k+1} = \sum_{i=0}^{k} \boldsymbol{\Lambda}^i \mathbf{P}^{-1}\mathbf{U}_{k+1-i}\mathbf{B}\mathbf{V}\boldsymbol{\Sigma}^i, \tag{24}$$

$$\mathbf{Y}_{k+1} = \phi(\mathbf{P}\mathbf{Z}_{k+1}\mathbf{V}^{-1}\mathbf{W}_1)\mathbf{W}_2, \tag{25}$$

Equations (24) and (25) give the same exact dynamics of the equations (17) and (18).

The matrix of complex eigenvectors $\mathbf{V}$ in (24) can be merged into the real matrix of weights $\mathbf{B}$ in equation (26). Therefore, we can call $\hat{\mathbf{B}}$ a complex matrix of weights that accounts for the term $\mathbf{B}\mathbf{V}$. Similarly, the matrix eigenvectors $\mathbf{V}^{-1}$ in (25) can be merged into the matrix of weights $\mathbf{W}_1$ in equation (27), that we call $\hat{\mathbf{W}}_1$. To get an exact equivalence, we should exactly multiply by $\mathbf{V}$ and $\mathbf{V}^{-1}$, but merging these into learnable complex-valued matrices $\hat{\mathbf{B}}$ and $\hat{\mathbf{W}}_1$ then we get similar performance.

With these new notations, we can write the equivalent diagonalised complex-valued MP-SSM block:

$$\mathbf{Z}_{k+1} = \sum_{i=0}^{k} \boldsymbol{\Lambda}^i \hat{\mathbf{U}}_{k+1-i}\hat{\mathbf{B}}\boldsymbol{\Sigma}^i, \tag{26}$$

$$\mathbf{Y}_{k+1} = \phi(\mathbf{P}\mathbf{Z}_{k+1}\hat{\mathbf{W}}_1)\mathbf{W}_2, \tag{27}$$

where, in summary:

    • input is pre-processed as $\hat{\mathbf{U}}_{k+1-i} = \mathbf{P}^{-1}\mathbf{U}_{k+1-i}$,

    • $\mathbf{\Lambda}$ is the diagonal matrix of the eigenvalues of the GSO,

    • learnable recurrent weights are $\hat{\mathbf{B}}$ (complex and dense), and $\mathbf{\Sigma}$ (complex and diagonal)

    • learnable readout weights are $\hat{\mathbf{W}}_1$ (complex and dense), and $\mathbf{W}_2$ (real and dense)

 Equations (26)-(27) tell us that we can implement the whole recurrence efficiently in a closed-form
 solution that only involves powers of diagonal matrices.

 We provide in Algorithm 1, the pytorch-like implementation of the fast MP-SSM, provided the input
 sequence $(\hat{\mathbf{U}}_1, \ldots, \hat{\mathbf{U}}_{k+1})$, computes in parallel the whole output sequence $(\mathbf{Y}_1, \ldots, \mathbf{Y}_{k+1})$.

---

**Algorithm 1** MP-SSM fast implementation

---

**Require:** the input features $\mathtt{x} \in \mathbb{C}^{\text{num\_steps} \times n \times C}$ (if temporal), else $\mathtt{x} \in \mathbb{C}^{n \times C}$; the number of
iterations (i.e., k+1) $\mathtt{num\_steps}$; the diagonal complex-valued weight matrix $\mathtt{W} \in \mathbb{C}^{\text{hidden\_dim}}$; the
complex-valued matrix $\mathtt{B} \in \mathbb{C}^{C \times \text{hidden\_dim}}$; the eigenvalues of the GSO $\mathtt{eigenvals} \in \mathbb{C}^n$
**Ensure:** $\mathtt{out} \in \mathbb{C}^{\text{num\_steps} \times n \times \text{hidden\_dim}}$

1: $\text{powers} = \text{torch.arange}(\mathtt{num\_steps})$
2: $\Lambda_{\text{powers}} = \mathtt{eigenvals}.\text{unsqueeze}(-1).\text{pow}(\text{powers})$ ▷ shape: $(n, \text{num\_steps})$
3: $\Sigma_{\text{powers}} = \mathtt{W}.\text{unsqueeze}(-1).\text{pow}(\text{powers})$ ▷ shape: (hidden\_dim, num\_steps)
4: **if not** temporal **then**
5:    $\mathtt{x} = \mathtt{x}.\text{repeat}(\mathtt{num\_steps}, 1, 1)$ ▷ shape: (num\_steps, $n$, $C$), static case
6: **end if**
7: $\mathtt{x}_{\text{flipped}} = \text{torch.flip}(\mathtt{x}, \text{dims} = [0])$ ▷ shape: (num\_steps, $n$, $C$)
8: $\mathtt{x}_{\text{complex}} = \mathtt{x}_{\text{flipped}}.\text{to}(\text{torch.cfloat})$
9: $\mathtt{x}_B = \text{torch.matmul}(\mathtt{x}_{\text{complex}}, \mathtt{B})$ ▷ shape: (num\_steps, $n$, hidden\_dim)
10: $\Lambda_{\text{powers}} = \Lambda_{\text{powers}}.\text{permute}(2, 0, 1)$ ▷ shape: (num\_steps, $n$, 1)
11: $\Sigma_{\text{powers}} = \Sigma_{\text{powers}}.\text{transpose}(1, 0).\text{unsqueeze}(1)$ ▷ shape: (num\_steps, 1, hidden\_dim)
12: $\text{scaled\_x\_B} = \Lambda_{\text{powers}} \cdot \mathtt{x}_B \cdot \Sigma_{\text{powers}}$
13: $\mathtt{out} = \text{scaled\_x\_B}.\text{cumsum}(\text{dim} = 0)$ ▷ shape: (num\_steps, $n$, hidden\_dim)
14: $\mathtt{d}_1, \mathtt{d}_2, \mathtt{d}_3 = \mathtt{out}.\text{shape}$
15: $\mathtt{x}_{\text{agg}} = \mathtt{out}.\text{permute}(1, 2, 0).\text{reshape}(n, -1)$ ▷ shape: ($n$, num\_steps · hidden\_dim)
16: $\mathtt{x}_{\text{agg}} = \text{matmul}($
        $\mathtt{x} = \mathtt{x}_{\text{agg}},$
        $\text{edge\_index} = \text{matrix\_p\_edge\_index},$
        $\text{edge\_weight} = \text{matrix\_p\_edge\_weight}$
    $)$
17: $\mathtt{x}_{\text{agg}} = \mathtt{x}_{\text{agg}}.\text{reshape}(\mathtt{d}_2, \mathtt{d}_3, \mathtt{d}_1).\text{permute}(2, 0, 1)$
18: $\mathtt{out} = \text{mlp}(\mathtt{x}_{\text{agg}}, \text{batch})$

---

 We acknowledge that there is no free lunch: we achieve a one-shot parallel implementation trading
 off GPU memory usage, since the whole tensor of shape (num\_steps, $n$, hidden\_dim), in line 9 of
 Algorithm 1, must fit into the GPU. However, with sufficient GPU memory, the entire MP-SSM block
 computation occurs in $10^{-3}$ seconds, see Figure 3. As shown in Figure 3, MP-SSM scales similarly
 to GCN and GCN (weight sharing), whose lines are overlapping, but it is slightly faster, owing to the
 lack of nonlinearity in the recurrence—a benefit that grows with more iterations. On the other hand,
 the fast implementation of MP-SSM maintains constant runtime, provided enough GPU memory.

 Finally, we note that, unlike standard SSM models such as S4 and Mamba, which follow a Single-
 Input-Single-Output strategy—computing a separate SSM for each input channel and then mixing
 the results—our implementation in Algorithm 1 adopts a Multiple-Input-Multiple-Output strategy,
 enabling native handling of multivariate inputs.

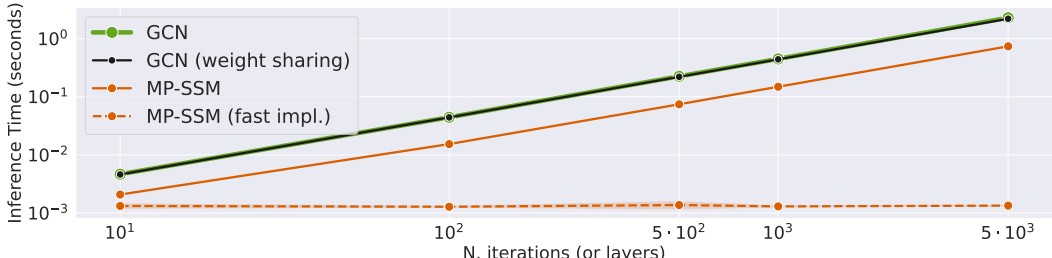

Figure 3: Inference time on a graph of $n = 100$ nodes (with number of edges 3058), input dimension $C = 1$, hidden_dim $= 32$, and increasing lengths $k = 10, 100, 500, 1000, 5000$. GCN is a standard GCN with $\tanh$ without residual with $k$ layers. GCN (weight sharing) is the same, but just one layer iterated $k$ times. MP-SSM baselines use both 1 block.

## F Relation to other temporal graph models based on state-space modeling

In the recent literature, we can find temporal graph models that leverage the state-space approach. The MP-SSM presents a simplified yet effective recurrent architecture for temporal graph modeling, offering clear advantages in architectural design when compared to alternatives such as GGRNN [90] or GraphSSM [66]. The MP-SSM recurrent dynamics are governed by a simple linear diffusion on the graph:

$$\mathbf{X}_{t+1} = \mathbf{A}\mathbf{X}_t\mathbf{W} + \mathbf{U}_{t+1}\mathbf{B}. \tag{28}$$

In contrast, the GGRNN recurrent equation (in its simplest form, without gating mechanisms) adopts a more elaborate design:

$$\mathbf{X}_{t+1} = \sigma\left(\sum_{j=0}^{K-1} \mathbf{A}^j\mathbf{X}_t\mathbf{W}_j + \sum_{j=0}^{K-1} \mathbf{A}^j\mathbf{U}_{t+1}\mathbf{B}_j\right), \tag{29}$$

where multiple powers of the shift operator, $\mathbf{A}$, are used to aggregate information from both previous embedding $\mathbf{X}_t$ and current input features $\mathbf{U}_{t+1}$, weighted with several learnable matrices, $\mathbf{W}_j$ and $\mathbf{B}_j$, which are applied for different $j$ values, and finally, applying a nonlinearity *at each time step*.

The key distinguishing feature of MP-SSM is the *absence of nonlinearity in the recurrent update*, with the only nonlinear transformation appearing in a downstream MLP decoder, typically composed of two dense layers with an activation function in between. This feature also allows for a fast implementation of the recurrence, since it can be unfolded to get a closed-form solution, see Appendix E. Moreover, in an MP-SSM block, the same weights, $\mathbf{W}, \mathbf{B}$ and MLP parameters, are shared across all time steps, ensuring *strict weight sharing throughout the sequence*. Moreover, our methodology implements a stack of MP-SSM blocks to build richer representations, differently from GGRNN where only one layer of recurrent computation is performed.

On the other hand, the GraphSSM model [66] adopts a strategy of stacking several GraphSSM blocks similar to MP-SSM, but their building blocks are fundamentally different from our MP-SSM block. In fact, a GraphSSM block processes the spatio-temporal input sequence $[\mathbf{U}_t]$ in three main stages, see Appendix D.2 of [66]. First, a GNN backbone is applied to the input sequence, generating a corresponding sequence of node embeddings $\mathbf{X}_t$. Next, each embedding is mixed with the one from the previous time step $\mathbf{X}_{t-1}$, producing a smoothed temporal embedding $\mathbf{H}_t$. This mixed sequence $[\mathbf{H}_t]$ is then treated as a multivariate time series and passed through an SSM layer—such as S4, S5, or S6—to yield the final sequence $[\mathbf{Y}_t]$ as the output of a GraphSSM block. Our approach is conceptually simpler, as it integrates both the GNN diffusive dynamics and sequence-based processing within a unified linear recurrence—Equation (28)—followed by a shared MLP applied across time steps. In this sense, MP-SSM embeds the core principles behind modern SSMs—the very principles that have driven the success of sequential modeling—directly into the graph processing framework. In contrast, GraphSSM merely combines GNN and SSM backbones in a modular fashion to address temporal graph tasks, without deeply integrating their underlying mechanisms.

In Table 3, we provide a direct comparison between MP-SSM, GGRNN, and GraphSSM, on the Metr-LA and PeMS-Bay datasets. To ensure a fair and comprehensive comparison, we computed MAE,

RMSE, and MAPE for all three models: MP-SSM, GGRNN, and GraphSSM. We used GGRNN without gating mechanisms, as it achieved the best performance on Metr-LA according to [90, Table IV], and GraphSSM-S4, since the authors reported in [66] that their experiments were primarily conducted using the S4 architecture. As the results show, our method consistently and significantly outperforms both GGRNN and GraphSSM across all three metrics on both datasets.

Table 3: Multivariate time series forecasting on the Metr-LA and PeMS-Bay datasets for Horizon 12. **Best** results for each task are in bold.

| Model | Metr-LA | | | PeMS-Bay | | |
|---|---|---|---|---|---|---|
| | MAE ↓ | RMSE ↓ | MAPE ↓ | MAE ↓ | RMSE ↓ | MAPE ↓ |
| GGRNN | 3.88 | 8.14 | 10.59% | 2.34 | 5.14 | 5.21% |
| GraphSSM-S4 | 3.74 | 7.90 | 10.37% | 1.98 | 4.45 | 4.77% |
| MP-SSM (ours) | **3.17** | **6.86** | **9.21%** | **1.62** | **4.22** | **4.05%** |

# G   Multi-hop interpretation of a deep MP-SSM architecture

MP-SSM is fundamentally different from multi-hop GNNs approaches: it operates through strictly 1-hop message passing at each iteration and does not perform aggregation from far-away hops by design. Nonetheless, to better understand its behavior in deeper architectures, we explore how a multi-hop perspective can be used for interpretation, drawing contrasts with a representative multi-hop model, Drew [49]. For this purpose, let us consider the static case, with the input being the sequence $[\mathbf{U}_1, \ldots, \mathbf{U}_1]$. The linearity of the recurrent equation of an MP-SSM block allows us to unfold the recurrent equation as follows:

$$\mathbf{X}_{k+1} = \mathbf{A}^{k+1}\mathbf{X}_0\mathbf{W}^{k+1} + \sum_{i=0}^{k} \mathbf{A}^i\mathbf{U}_1\mathbf{B}\mathbf{W}^i. \tag{30}$$

Therefore, assuming a zero initial state and including the MLP into the equation, we have the following expression in the output of the first MP-SSM block:

$$\mathbf{Y}_{k+1} = \text{MLP}\Big(\sum_{i=0}^{k} \mathbf{A}^i\mathbf{U}_1\mathbf{B}\mathbf{W}^i\Big). \tag{31}$$

Due to the various powers of the shift operator $\mathbf{I}, \mathbf{A}, \mathbf{A}^2, \ldots, \mathbf{A}^k$, we can interpret Equation (31) as a $k$-hop aggregation of the input graph $\mathbf{U}_1$. Now, the sequence $[\mathbf{Y}_{k+1}, \ldots, \mathbf{Y}_{k+1}]$ is the input to the second MP-SSM block. Therefore, stacking the second MP-SSM block, and considering a residual connection from the first MP-SSM block, we have the following expression in the output of the second MP-SSM block:

$$\mathbf{Y}_{2(k+1)} = \mathbf{Y}_{k+1} + \text{MLP}\Big(\sum_{i=0}^{k} \mathbf{A}^i\mathbf{Y}_{k+1}\mathbf{B}_2\mathbf{W}_2^i\Big), \tag{32}$$

where $\mathbf{B}_2, \mathbf{W}_2$, are the shared weights of the second MP-SSM block. In general, in a deep MP-SSM architecture of $s$ blocks, we have the following expression in the output of the $s$-th MP-SSM block:

$$\mathbf{Y}_{s(k+1)} = \mathbf{Y}_{(s-1)(k+1)} + \text{MLP}\Big(\sum_{i=0}^{k} \mathbf{A}^i\mathbf{Y}_{(s-1)(k+1)}\mathbf{B}_s\mathbf{W}_s^i\Big). \tag{33}$$

To reveal the multi-hop view, we denote $\hat{\mathbf{Y}}^{(s)} = \mathbf{Y}_{s(k+1)}$, $\hat{\mathbf{W}}_i^{(s)} = \mathbf{B}_s\mathbf{W}_s^i$, and describe the deep MP-SSM architecture at the granularity of its blocks, as follows:

$$\hat{\mathbf{Y}}^{(s)} = \hat{\mathbf{Y}}^{(s-1)} + \text{MLP}\Big(\sum_{i=0}^{k} \mathbf{A}^i\hat{\mathbf{Y}}^{(s-1)}\hat{\mathbf{W}}_i^{(s)}\Big). \tag{34}$$

This multi-hop interpretation of a deep MP-SSM architecture resembles the DRew-GCN architecture [49], a multi-hop MPNN employing a dynamically rewired message passing strategy with delay. In

fact, the recurrent equation of DRew-GCN, rephrased in our MP-SSM notation for ease of comparison, is defined as:

$$\mathbf{Y}^{(s+1)} = \mathbf{Y}^{(s)} + \sigma \left( \sum_{i=1}^{s+1} \mathbf{A}(i)\mathbf{Y}^{(s-\tau_\nu(i))}\mathbf{W}_i^{(s)} \right), \tag{35}$$

where $\mathbf{A}(i)$ is the degree-normalised shift operator that considers all the neighbors at an *exact* $i$ hops from each respective root node, $\mathbf{W}_i^{(s)}$ are weight matrices, and $\tau_\nu(i)$ is a positive integer (the *delay*) defining the temporal window for the aggregation of past embeddings. Comparing Equation (34) and Equation (35) we can summarize the following differences:

- DRew aggregates information using $\mathbf{A}(i)$, a function of the GSO that counts neighbors at an *exact* $i$ hops distance, while MP-SSM considers the powers of the GSO, $\mathbf{A}^i$, thus accounting for all the possible walks of length $i$. Similarly, the learnable weights in MP-SSM reflect the architectural bias induced by the recurrence, as they are structured through powers of a base matrix, specifically following the form $\hat{\mathbf{W}}_i^{(s)} = \mathbf{B}_s \mathbf{W}_s^i$.

- DRew nonlinearly aggregates information via a pointwise nonlinearity $\sigma$, while MP-SSM employs a more expressive 2-layers MLP.

- MP-SSM uses the same features for multi-hop aggregation (corresponding to $\tau_\nu(i) \equiv 0$), whereas DRew aggregates features from previous layers with a delay $\tau_\nu(i) = \max(0, i - \nu)$, effectively introducing a temporal rewiring of the graph.

Although the unfolding of MP-SSM yields expressions involving powers of the GSO, this resemblance to multi-hop architectures such as DRew [49] is purely superficial. Unlike models that aggregate information from distant nodes within a single layer, MP-SSM performs strictly 1-hop message passing at each iteration. The higher-order GSO terms emerge naturally from the recurrence, not from an architectural bias toward multi-hop aggregation. This formulation, grounded in first principles, preserves the original graph topology and constitutes a structurally distinct approach. We provide in Table 4 a comparison of DRew-GCN (results taken from [49]) with our MP-SSM on the Peptides-func and Peptides-struct from the LRGB task [32]. Notably, MP-SSM outperforms DRew-GCN on the Peptides-struct task, suggesting that the structural architectural bias introduced by the recurrence, combined with MLP adaptivity, offers a stronger advantage than aggregating information via rewired connections from delayed past features. In contrast, on the Peptides-func task, the performance of the two models falls within each other's standard deviation, indicating no statistically significant difference between DRew-GCN—despite its dynamic rewiring strategy with delay—and MP-SSM. In Appendix M we report an extended evaluation on the LRGB benchmark.

Table 4: Results for Peptides-func and Peptides-struct averaged over 3 training seeds. DRew-GCN results are taken from [49]. The **best** scores are in bold.

| Model | Peptides-func AP ↑ | Peptides-struct MAE ↓ |
|---|---|---|
| DRew-GCN | **69.96**$_{\pm 0.76}$ | 0.2781$_{\pm 0.0028}$ |
| MP-SSM (ours) | 69.93$_{\pm 0.52}$ | **0.2458**$_{\pm 0.0017}$ |

# H Ablations

We perform an ablation study to isolate the incremental contribution of each SSM heuristic to the performance gains in reconstructing graph-structural information that depends on learning long-range dependencies; specifically for computing quantities like the diameter of a graph, the single-source-shortest-paths (SSSP), and the eccentricity of a node, see Section 4.1 for more details on these tasks. Results of this ablation are reported in Table 5.

The ablation conducted reveals that removing the nonlinearity from GCN yields the most significant performance improvement. Introducing weight sharing—effectively incorporating recurrence into the linear graph diffusion process—yields a slight performance boost while considerably reducing the number of parameters. Appending an MLP at the last time step of this linear recurrent architecture does not result in statistically significant gains, except marginally for the Eccentricity task. Likewise,

Table 5: Architecture ablation study. Mean test $log_{10}(MSE)$ and std averaged on 4 random weight initialization on Graph Property Prediction tasks (Section 4.1). The lower, the better. The evaluation include: a nonlinear multilayer GCN (`GCN`), a linear multilayer GCN (`Linear GCN`), a linear multilayer GCN with weight sharing (`Linear GCN (ws)`), Linear GCN (ws) followed by an MLP (`1 Block Linear GCN`), a stack of multiple 1 Block Linear GCN (`Multi-Blocks Linear GCN`), and our `MP-SSM`, which represent a multi-blocks linear GCN with standard deep learning heuristics such as residual connections and normalisation layers between blocks.

| Model | Diameter ↓ | SSSP ↓ | Eccentricity ↓ |
|---|---|---|---|
| GCN | $0.7424_{\pm 0.0466}$ | $0.9499_{\pm 0.0001}$ | $0.8468_{\pm 0.0028}$ |
| Linear GCN | $-2.1255_{\pm 0.0984}$ | $-1.5822_{\pm 0.0002}$ | $-2.1424_{\pm 0.0014}$ |
| Linear GCN (ws) | $-2.2678_{\pm 0.1277}$ | $-1.5823_{\pm 0.0001}$ | $-2.1447_{\pm 0.001}$ |
| 1 Block Linear GCN | $-2.2734_{\pm 0.1513}$ | $-1.5836_{\pm 0.0025}$ | $-2.1869_{\pm 0.0058}$ |
| Multi-Blocks Linear GCN | $-2.3531_{\pm 0.3183}$ | $-1.5821_{\pm 0.0001}$ | $-2.1861_{\pm 0.0066}$ |
| MP-SSM | $\mathbf{-3.2353}_{\pm 0.1735}$ | $\mathbf{-4.6321}_{\pm 0.0779}$ | $\mathbf{-2.9724}_{\pm 0.0271}$ |

constructing a hierarchical block structure does not noticeably enhance performance. These limited improvements suggest that, for the three tasks considered, the linear recurrence mechanism alone, provided a long enough recurrence, is sufficient to capture meaningful representations to reconstruct graph's structural information. Finally, incorporating standard deep learning heuristics further strengthens the full MP-SSM architecture, consistently improving performance across all tasks.

# I   Complexity and Runtimes

We discuss the theoretical complexity of our method, followed by a comparison of runtimes with other methods.

**Complexity Analysis.** Our MP-SSM consists of a stack of blocks. Each of them performs a linear recurrence of $k$ iterations followed by the application of a nonlinear map, as defined in Equations (1) and (2). Note that $k$ is either the length of the temporal graph sequence or a hyperparameter. Given the similarities between the linear recurrence in MP-SSM and standard MPNNs, described in Section 2, the recurrence retains the complexity of standard MPNNs. Therefore, the Equation (1) is linear in the number of node $|V|$ and edges $|E|$, achieving a time complexity of $\mathcal{O}(k \cdot (|V| + |E|))$, with $k$ the number of iterations. Considering $\mathcal{O}(m)$ the time complexity of the MLP in Equation (2), then the final time complexity of one MP-SSM block is $\mathcal{O}(k \cdot (|V| + |E|) + m)$ in the static case and $\mathcal{O}(k \cdot (|V| + |E| + m))$ in the temporal case.

**Runtimes.** We provide runtimes for MP-SSM and compare it with other methods, such as Graph GPS and GCN, in Table 6. In all cases, we use a model with 256 hidden dimensions and a varying depth effective by changing the number of recurrences from 2 to 16 in our MP-SSM with 2 MP-SSM blocks, and the number of layers is the depth for other methods. We report the training and inference times in milliseconds, as well as the downstream performance performance obtained on the Roman-Empire dataset. As can be seen from the results in the Table, our MP-SSM maintains a similar runtime to GCN, which has linear complexity with respect to the graph size, while offering strong performance at the same time. Notably, our MP-SSM achieves better performance than GCN and GPS, and maintains its performance as depth increases, different than GCN. All runtimes are measured on an NVIDIA A6000 GPU with 48GB of memory.

# J   The vanishing gradient tendency in nonlinear MPNNs.

Let us consider a highly connected graph without bottlenecks, such that the transfer of messages from any node to any other node is not affected by issues due to structural properties of the graph. However, in the deep regime, the presence of a nonlinearity at each time step can lead the global sensitivity (as defined in Equation (9)) to be vanishing small.

For an MP-SSM block, the local sensitivity $\mathcal{S}_{ij}(t-s)$ of the features of the $i$-th node to features of the $j$-th node after $t-s$ applications of message-passing aggregations, is exactly the norm of the Jacobian of Equation (6), i.e. the norm of the product of the $(i,j)$-entry of $\mathbf{A}^{t-s}$ and the matrix $(\mathbf{W}^\top)^{t-s}$. For standard MPNN approaches, the local sensitivity has a more complicated expression due to

Table 6: Training and Inference Runtime (milliseconds) and obtained node classification accuracy (%) on the Roman-Empire dataset.

| Metrics | Method | Depth | | | |
|---|---|---|---|---|---|
| | | 4 | 8 | 16 | 32 |
| Training (ms) | | 18.38 | 33.09 | 61.86 | 120.93 |
| Inference (ms) | GCN | 9.30 | 14.64 | 27.95 | 53.55 |
| Accuracy (%) | | 73.60 | 61.52 | 56.86 | 52.42 |
| Training (ms) | | 1139.05 | 2286.96 | 4545.46 | OOM |
| Inference (ms) | GPS | 119.10 | 208.26 | 427.89 | OOM |
| Accuracy (%) | | 81.97 | 81.53 | 81.88 | OOM |
| Training (ms) | | 1179.08 | 2304.77 | 4590.26 | OOM |
| Inference (ms) | GPSGAT+Performer (RWSE) | 120.11 | 209.98 | 429.03 | OOM |
| Accuracy (%) | | 84.89 | 87.01 | 86.94 | OOM |
| Training (ms) | | 23.19 | 41.44 | 72.09 | 141.82 |
| Inference (ms) | MP-SSM | 10.93 | 18.87 | 38.87 | 67.59 |
| Accuracy (%) | | 85.73 | 88.02 | 90.82 | 90.91 |

nonlinearities at each aggregation step, but usually there are 3 key contributors: one from several multiplications of the shift operator (akin to $\mathbf{A}^{t-s}$ in our MP-SSM), one from several multiplications of the weights (akin to $(\mathbf{W}^\top)^{t-s}$ in our MP-SSM), and one from several multiplications of the derivative of the nonlinearity evaluated on the sequence of embeddings $\mathbf{D}(s), \mathbf{D}(s+1), .., \mathbf{D}(t)$. Usually the nonlinearity is pointwise, so $\mathbf{D}(t)$ is a diagonal matrix with entries usually in $[0, 1]$, thus contributing to vanishing the gradient more and more at each time step. Hence, if the subsequent multiplications of weights and nonlinearity-based terms tend to vanish, while the powers of the shift operator $\mathbf{A}$ are bounded (as it is for the case of the symmetrically normalized adjacency with self-loops, proved in Lemma 4.5) then the local sensitivity tends to vanish *for all pair of nodes*, for $t - s$ large enough. This will be reflected in the global sensitivity, which also will tend to vanish, for $t - s$ large enough. This demonstrates that global sensitivity effectively quantifies the severity of vanishing gradient issues in MPNN models plagued by this problem. Note further that the local sensitivity of the linear recurrence in each block of our MP-SSM has the exact form of $||(\mathbf{A}^{t-s})_{ij}(\mathbf{W}^\top)^{t-s}||$, while for standard MPNN approaches with nonlinearities at each time step the vanishing effect will be stronger, as we formally proved for the case of GCN in Theorem C.13.

## K  Extended comparison on the Graph Property Prediciton Benchmark

To further evaluate the performance of MP-SSM, we report a more complete comparison for the GPP task in Table 7. Specifically, we include more MPNN-based models.

## L  Further spatio-temporal benchmarks

In Table 8, we report the results for Chickenpox Hungary, PedalMe London, and Wikipedia math [89], which involve public health, delivery demand, and web activity.

As evident from the table, MP-SSM achieves the best results across all datasets.

## M  Results on the Long-Range Graph Benchmark.

To further evaluate the performance of our MP-SSM, we consider two tasks of the Long-Range Graph Benchmark (LRGB) [32].

**Setup.** We evaluate MP-SSM on the Peptides-func and Peptides-struct tasks from the LRGB benchmark, which involve predicting functional and structural properties of peptides that require modeling long-range dependencies. We follow the original experimental setup and 500k parameter budget.

Table 7: Mean test set $log_{10}(\text{MSE})(\downarrow)$ and std averaged on 4 random weight initializations on Graph Property Prediction tasks. The lower the better. **First**, **second**, and **third** best results for each task are color-coded.

| Model | Diameter | SSSP | Eccentricity |
|---|---|---|---|
| **MPNNs** | | | |
| A-DGN | $-0.5188_{\pm 0.1812}$ | $-3.2417_{\pm 0.0751}$ | $0.4296_{\pm 0.1003}$ |
| DGC | $0.6028_{\pm 0.0050}$ | $-0.1483_{\pm 0.0231}$ | $0.8261_{\pm 0.0032}$ |
| GAT | $0.8221_{\pm 0.0752}$ | $0.6951_{\pm 0.1499}$ | $0.7909_{\pm 0.0222}$ |
| GCN | $0.7424_{\pm 0.0466}$ | $0.9499_{\pm 0.0001}$ | $0.8468_{\pm 0.0028}$ |
| GCNII | $0.5287_{\pm 0.0570}$ | $-1.1329_{\pm 0.0135}$ | $0.7640_{\pm 0.0355}$ |
| GIN | $0.6131_{\pm 0.0990}$ | $-0.5408_{\pm 0.4193}$ | $0.9504_{\pm 0.0007}$ |
| GRAND | $0.6715_{\pm 0.0490}$ | $-0.0942_{\pm 0.3897}$ | $0.6602_{\pm 0.1393}$ |
| GraphCON | $0.0964_{\pm 0.0620}$ | $-1.3836_{\pm 0.0092}$ | $0.6833_{\pm 0.0074}$ |
| GraphSAGE | $0.8645_{\pm 0.0401}$ | $0.2863_{\pm 0.1843}$ | $0.7863_{\pm 0.0207}$ |
| **Transformers** | | | |
| GPS | $-0.5121_{\pm 0.0426}$ | $-3.5990_{\pm 0.1949}$ | $0.6077_{\pm 0.0282}$ |
| **Ours** | | | |
| MP-SSM | $-3.2353_{\pm 0.1735}$ | $-4.6321_{\pm 0.0779}$ | $-2.9724_{\pm 0.0271}$ |

Table 8: Average MSE and standard deviation ($\downarrow$) of 10 experimental repetitions. Baseline results are reported from [89, 35, 34] . **First**, **second**, and **third** best methods for each task are color-coded.

| Model | Chickenpox Hungary | PedalMe London | Wikipedia Math |
|---|---|---|---|
| **Temporal GNNs** | | | |
| A3T-GCN | $1.114_{\pm 0.008}$ | $1.469_{\pm 0.027}$ | $0.781_{\pm 0.011}$ |
| AGCRN | $1.120_{\pm 0.010}$ | $1.469_{\pm 0.030}$ | $0.788_{\pm 0.011}$ |
| CDE | $0.848_{\pm 0.020}$ | $0.810_{\pm 0.063}$ | $0.694_{\pm 0.028}$ |
| DCRNN | $1.124_{\pm 0.015}$ | $1.463_{\pm 0.019}$ | $0.679_{\pm 0.020}$ |
| DyGrAE | $1.120_{\pm 0.021}$ | $1.455_{\pm 0.031}$ | $0.773_{\pm 0.009}$ |
| DynGESN | $0.907_{\pm 0.007}$ | $1.528_{\pm 0.063}$ | $0.610_{\pm 0.003}$ |
| EGCN-O | $1.124_{\pm 0.009}$ | $1.491_{\pm 0.024}$ | $0.750_{\pm 0.014}$ |
| GConvGRU | $1.128_{\pm 0.011}$ | $1.622_{\pm 0.032}$ | $0.657_{\pm 0.015}$ |
| GC-LSTM | $1.115_{\pm 0.014}$ | $1.455_{\pm 0.023}$ | $0.779_{\pm 0.023}$ |
| GRAND | $1.068_{\pm 0.021}$ | $1.557_{\pm 0.049}$ | $0.798_{\pm 0.034}$ |
| GREAD | $0.983_{\pm 0.027}$ | $1.291_{\pm 0.055}$ | $0.704_{\pm 0.016}$ |
| HMM4G | $0.939_{\pm 0.013}$ | $1.769_{\pm 0.370}$ | $0.542_{\pm 0.008}$ |
| MPNN LSTM | $1.116_{\pm 0.023}$ | $1.485_{\pm 0.028}$ | $0.795_{\pm 0.010}$ |
| TDE-GNN | $0.787_{\pm 0.018}$ | $0.714_{\pm 0.051}$ | $0.565_{\pm 0.017}$ |
| T-GCN | $1.117_{\pm 0.011}$ | $1.479_{\pm 0.012}$ | $0.764_{\pm 0.011}$ |
| **Ours** | | | |
| MP-SSM | $0.748_{\pm 0.011}$ | $0.647_{\pm 0.062}$ | $0.509_{\pm 0.008}$ |

**Results.** As shown in Table 9, MP-SSM outperforms standard MPNNs, transformer-based GNNs, and most multi-hop and SSM-based models. It achieves the highest average ranking across tasks without relying on global attention or graph rewiring. Compared to other graph SSMs, MP-SSM delivers strong performance while preserving permutation-equivariance.

# N   Results on the Heterophilic Benchmark

To further evaluate the performance of MP-SSM, we report a thorough comparison for the heterophilic task in Table 10. Specifically, we include many MPNN-based models, graph transformers, and heterophily-designated GNNs.

In Table 10, we color the top three methods. Notably, our MP-SSM achieves the best average ranking across all datasets in the heterophilic benchmarks.

Table 9: Results for Peptides-func and Peptides-struct averaged over 3 training seeds. Re-evaluated methods employ the 3-layer MLP readout proposed in [105]. Note that all MPNN-based methods include structural and positional encoding. The **first**, **second**, and **third** best scores are colored. Baseline results are reported from [32, 49, 105, 53, 28, 44]. ‡ means 3-layer MLP readout and residual connections are employed.

| Model | Peptides-func AP ↑ | Peptides-struct MAE ↓ | avg. Rank ↓ |
|---|---|---|---|
| **MPNNs** | | | |
| A-DGN | $59.75_{\pm 0.44}$ | $0.2874_{\pm 0.0021}$ | 26.0 |
| GatedGCN | $58.64_{\pm 0.77}$ | $0.3420_{\pm 0.0013}$ | 29.0 |
| GCN | $59.30_{\pm 0.23}$ | $0.3496_{\pm 0.0013}$ | 29.5 |
| GCNII | $55.43_{\pm 0.78}$ | $0.3471_{\pm 0.0010}$ | 30.5 |
| GINE | $54.98_{\pm 0.79}$ | $0.3547_{\pm 0.0045}$ | 32.0 |
| GRAND | $57.89_{\pm 0.62}$ | $0.3418_{\pm 0.0015}$ | 29.0 |
| GraphCON | $60.22_{\pm 0.68}$ | $0.2778_{\pm 0.0018}$ | 24.0 |
| SWAN | $67.51_{\pm 0.39}$ | $0.2485_{\pm 0.0009}$ | 12.5 |
| **Multi-hop GNNs** | | | |
| DIGL+MPNN | $64.69_{\pm 0.19}$ | $0.3173_{\pm 0.0007}$ | 25.0 |
| DIGL+MPNN+LapPE | $68.30_{\pm 0.26}$ | $0.2616_{\pm 0.0018}$ | 16.5 |
| DRew-GatedGCN | $67.33_{\pm 0.94}$ | $0.2699_{\pm 0.0018}$ | 19.5 |
| DRew-GatedGCN+LapPE | $69.77_{\pm 0.26}$ | $0.2539_{\pm 0.0007}$ | 12.0 |
| DRew-GCN | $69.96_{\pm 0.76}$ | $0.2781_{\pm 0.0028}$ | 14.0 |
| DRew-GCN+LapPE | $\mathbf{71.50}_{\pm 0.44}$ | $0.2536_{\pm 0.0015}$ | 8.0 |
| DRew-GIN | $69.40_{\pm 0.74}$ | $0.2799_{\pm 0.0016}$ | 17.5 |
| DRew-GIN+LapPE | $71.26_{\pm 0.45}$ | $0.2606_{\pm 0.0014}$ | 9.5 |
| GRED | $\mathbf{70.85}_{\pm 0.27}$ | $0.2503_{\pm 0.0019}$ | 7.0 |
| MixHop-GCN | $65.92_{\pm 0.36}$ | $0.2921_{\pm 0.0023}$ | 23.0 |
| MixHop-GCN+LapPE | $68.43_{\pm 0.49}$ | $0.2614_{\pm 0.0023}$ | 15.5 |
| **Transformers** | | | |
| GraphGPS+LapPE | $65.35_{\pm 0.41}$ | $0.2500_{\pm 0.0005}$ | 15.5 |
| Graph ViT | $69.42_{\pm 0.75}$ | $\mathbf{0.2449}_{\pm 0.0016}$ | 5.5 |
| GRIT | $69.88_{\pm 0.82}$ | $\mathbf{0.2460}_{\pm 0.0012}$ | **5.0** |
| Transformer+LapPE | $63.26_{\pm 1.26}$ | $0.2529_{\pm 0.0016}$ | 19.5 |
| SAN+LapPE | $63.84_{\pm 1.21}$ | $0.2683_{\pm 0.0043}$ | 22.0 |
| **Modified and Re-evaluated**‡ | | | |
| DRew-GCN+LapPE | $69.45_{\pm 0.21}$ | $0.2517_{\pm 0.0011}$ | 11.0 |
| GatedGCN | $67.65_{\pm 0.47}$ | $0.2477_{\pm 0.0009}$ | 11.0 |
| GCN | $68.60_{\pm 0.50}$ | $\mathbf{0.2460}_{\pm 0.0007}$ | 7.5 |
| GINE | $66.21_{\pm 0.67}$ | $0.2473_{\pm 0.0017}$ | 12.0 |
| GraphGPS+LapPE | $65.34_{\pm 0.91}$ | $0.2509_{\pm 0.0014}$ | 17.0 |
| **Graph SSMs** | | | |
| GMN | $70.71_{\pm 0.83}$ | $0.2473_{\pm 0.0025}$ | 4.5 |
| Graph-Mamba | $67.39_{\pm 0.87}$ | $0.2478_{\pm 0.0016}$ | 12.5 |
| **Ours** | | | |
| MP-SSM | $69.93_{\pm 0.52}$ | $\mathbf{0.2458}_{\pm 0.0017}$ | 4.0 |

# O   Experimental Details

## O.1   Employed baselines

In our experiments, the performance of our method is compared with various state-of-the-art GNN baselines from the literature. Specifically, we consider:

- classical MPNN-based methods, i.e., GCN [61], GraphSAGE [50], GAT [109], GatedGCN [13], GIN [118], ARMA [11], GINE [56], GCNII [18], and CoGNN [37];

- heterophily-specific models, i.e., H2GCN [127], CPGNN [126], FAGCN [12], GPR-GNN [19], FSGNN [74], GloGNN [67], GBK-GNN [29], and JacobiConv [113];

- physics-inspired MPNNs, i.e., DGC [114], GRAND [15], GraphCON [91], A-DGN [43], GREAD [20], CDE [123], and TDE-GNN [34];

- Graph Transformers, i.e., Transformer [107, 30], GT [95], SAN [63], GPS [88], GOAT [62], Exphormer [97], NAGphormer [16], GRIT [73], and GraphViT [53];

- Higher-Order DGNs, i.e., DIGL [40], MixHop [1], DRew [49], and GRED [28].

- SSM-based GNN, i.e., Graph-Mamba [111], GMN [10], GPS+Mamba [10], GGRNN [90], and GraphSSM [66].

Table 10: Mean test set score and std averaged over 4 random weight initializations on heterophilic datasets. The higher, the better. First, second, and **third** best results for each task are color-coded. Baseline results are reported from [37, 10, 86, 78, 72]. "∗" in the rank column means that the average has been computed over less trials.

| Model | Roman-empire Acc ↑ | Amazon-ratings Acc ↑ | Minesweeper AUC ↑ | Tolokers AUC ↑ | Questions AUC ↑ | avg. Rank ↓ |
|---|---|---|---|---|---|---|
| **[72]** | | | | | | |
| MLP-1 | $64.12_{\pm 0.61}$ | $38.60_{\pm 0.41}$ | $50.59_{\pm 0.83}$ | $71.89_{\pm 0.82}$ | $70.33_{\pm 0.96}$ | 41.0 |
| MLP-2 | $66.04_{\pm 0.71}$ | $49.55_{\pm 0.81}$ | $50.92_{\pm 1.25}$ | $74.58_{\pm 0.75}$ | $69.97_{\pm 1.16}$ | 34.4 |
| SGC-1 | $44.60_{\pm 0.52}$ | $40.69_{\pm 0.42}$ | $82.04_{\pm 0.77}$ | $73.80_{\pm 1.35}$ | $71.06_{\pm 0.92}$ | 38.6 |
| **Graph-agnostic** | | | | | | |
| ResNet | $65.88_{\pm 0.38}$ | $45.90_{\pm 0.52}$ | $50.89_{\pm 1.39}$ | $72.95_{\pm 1.06}$ | $70.34_{\pm 0.76}$ | 37.4 |
| ResNet+adj | $52.25_{\pm 0.40}$ | $51.83_{\pm 0.57}$ | $50.42_{\pm 0.83}$ | $78.78_{\pm 1.11}$ | $75.77_{\pm 1.24}$ | 32.0 |
| ResNet+SGC | $73.90_{\pm 0.51}$ | $50.66_{\pm 0.48}$ | $70.88_{\pm 0.90}$ | $80.70_{\pm 0.97}$ | $75.81_{\pm 0.96}$ | 29.0 |
| **MPNNs** | | | | | | |
| CO-GNN($\Sigma, \Sigma$) | $91.57_{\pm 0.32}$ | $51.28_{\pm 0.56}$ | $95.09_{\pm 1.18}$ | $83.36_{\pm 0.89}$ | $80.02_{\pm 0.86}$ | **8.0** |
| CO-GNN($\mu, \mu$) | $91.37_{\pm 0.35}$ | $54.17_{\pm 0.41}$ | $97.31_{\pm 0.67}$ | $84.45_{\pm 1.17}$ | $76.54_{\pm 0.95}$ | 6.8 |
| GAT | $80.87_{\pm 0.30}$ | $49.09_{\pm 0.63}$ | $92.01_{\pm 0.68}$ | $83.70_{\pm 0.47}$ | $77.43_{\pm 1.20}$ | 18.0 |
| GAT-sep | $88.75_{\pm 0.41}$ | $52.70_{\pm 0.62}$ | $93.91_{\pm 0.35}$ | $83.78_{\pm 0.43}$ | $76.79_{\pm 0.71}$ | 9.8 |
| GAT (LapPE) | $84.80_{\pm 0.46}$ | $44.90_{\pm 0.73}$ | $93.50_{\pm 0.54}$ | $84.99_{\pm 0.54}$ | $76.55_{\pm 0.84}$ | 16.0 |
| GAT (RWSE) | $86.62_{\pm 0.53}$ | $48.58_{\pm 0.41}$ | $92.53_{\pm 0.65}$ | $85.02_{\pm 0.67}$ | $77.83_{\pm 1.22}$ | 11.6 |
| GAT (DEG) | $85.51_{\pm 0.56}$ | $51.65_{\pm 0.60}$ | $93.04_{\pm 0.62}$ | $84.22_{\pm 0.81}$ | $77.10_{\pm 1.23}$ | 12.6 |
| Gated-GCN | $74.46_{\pm 0.54}$ | $43.00_{\pm 0.32}$ | $87.54_{\pm 1.22}$ | $77.31_{\pm 1.14}$ | $76.61_{\pm 1.13}$ | 31.4 |
| GCN | $73.69_{\pm 0.74}$ | $48.70_{\pm 0.63}$ | $89.75_{\pm 0.52}$ | $83.64_{\pm 0.67}$ | $76.09_{\pm 1.27}$ | 25.8 |
| GCN (LapPE) | $83.37_{\pm 0.55}$ | $44.35_{\pm 0.36}$ | $94.26_{\pm 0.49}$ | $84.95_{\pm 0.78}$ | $77.79_{\pm 1.34}$ | 14.6 |
| GCN (RWSE) | $84.84_{\pm 0.55}$ | $46.40_{\pm 0.55}$ | $93.84_{\pm 0.48}$ | $85.11_{\pm 0.77}$ | $77.81_{\pm 1.40}$ | 12.0 |
| GCN (DEG) | $84.21_{\pm 0.47}$ | $50.01_{\pm 0.69}$ | $94.14_{\pm 0.50}$ | $82.51_{\pm 0.83}$ | $76.96_{\pm 1.21}$ | 16.4 |
| SAGE | $85.74_{\pm 0.67}$ | $53.63_{\pm 0.39}$ | $93.51_{\pm 0.57}$ | $82.43_{\pm 0.44}$ | $76.44_{\pm 0.62}$ | 15.6 |
| **Graph Transformers** | | | | | | |
| Exphormer | $89.03_{\pm 0.37}$ | $53.51_{\pm 0.46}$ | $90.74_{\pm 0.53}$ | $83.77_{\pm 0.78}$ | $73.94_{\pm 1.06}$ | 16.6 |
| NAGphormer | $74.34_{\pm 0.77}$ | $51.26_{\pm 0.72}$ | $84.19_{\pm 0.66}$ | $78.32_{\pm 0.95}$ | $68.17_{\pm 1.53}$ | 30.6 |
| GOAT | $71.59_{\pm 1.25}$ | $44.61_{\pm 0.50}$ | $81.09_{\pm 1.02}$ | $83.11_{\pm 1.04}$ | $75.76_{\pm 1.66}$ | 31.2 |
| GPS | $82.00_{\pm 0.61}$ | $53.10_{\pm 0.42}$ | $90.63_{\pm 0.67}$ | $83.71_{\pm 0.48}$ | $71.73_{\pm 1.47}$ | 21.4 |
| GPS$_{GCN+Performer}$ (LapPE) | $83.96_{\pm 0.53}$ | $48.20_{\pm 0.67}$ | $93.85_{\pm 0.41}$ | $84.72_{\pm 0.77}$ | $77.85_{\pm 1.25}$ | 12.8 |
| GPS$_{GCN+Performer}$ (RWSE) | $84.72_{\pm 0.65}$ | $48.08_{\pm 0.85}$ | $92.88_{\pm 0.50}$ | $84.81_{\pm 0.86}$ | $76.45_{\pm 1.51}$ | 16.6 |
| GPS$_{GCN+Performer}$ (DEG) | $83.38_{\pm 0.68}$ | $48.93_{\pm 0.47}$ | $93.60_{\pm 0.47}$ | $80.49_{\pm 0.97}$ | $74.24_{\pm 1.18}$ | 22.6 |
| GPS$_{GAT+Performer}$ (LapPE) | $85.93_{\pm 0.52}$ | $48.86_{\pm 0.38}$ | $92.62_{\pm 0.79}$ | $84.62_{\pm 0.54}$ | $76.71_{\pm 0.98}$ | 14.4 |
| GPS$_{GAT+Performer}$ (RWSE) | $87.04_{\pm 0.58}$ | $49.92_{\pm 0.68}$ | $91.08_{\pm 0.58}$ | $84.38_{\pm 0.91}$ | $77.14_{\pm 1.49}$ | 15.0 |
| GPS$_{GAT+Performer}$ (DEG) | $85.54_{\pm 0.58}$ | $51.03_{\pm 0.60}$ | $91.52_{\pm 0.46}$ | $82.45_{\pm 0.89}$ | $76.51_{\pm 1.19}$ | 20.0 |
| GPS$_{GCN+Transformer}$ (LapPE) | OOM | OOM | $91.82_{\pm 0.41}$ | $83.51_{\pm 0.93}$ | OOM | 33.8 |
| GPS$_{GCN+Transformer}$ (RWSE) | OOM | OOM | $91.17_{\pm 0.51}$ | $83.53_{\pm 1.06}$ | OOM | 34.4 |
| GPS$_{GCN+Transformer}$ (DEG) | OOM | OOM | $91.76_{\pm 0.61}$ | $80.82_{\pm 0.95}$ | OOM | 36.2 |
| GPS$_{GAT+Transformer}$ (LapPE) | OOM | OOM | $92.29_{\pm 0.61}$ | $84.70_{\pm 0.56}$ | OOM | 30.2 |
| GPS$_{GAT+Transformer}$ (RWSE) | OOM | OOM | $90.82_{\pm 0.56}$ | $84.01_{\pm 0.96}$ | OOM | 33.8 |
| GPS$_{GAT+Transformer}$ (DEG) | OOM | OOM | $91.58_{\pm 0.56}$ | $81.89_{\pm 0.85}$ | OOM | 36.0 |
| GT | $86.51_{\pm 0.73}$ | $51.17_{\pm 0.66}$ | $91.85_{\pm 0.76}$ | $83.23_{\pm 0.64}$ | $77.95_{\pm 0.68}$ | 14.4 |
| GT-sep | $87.32_{\pm 0.39}$ | $52.18_{\pm 0.80}$ | $92.29_{\pm 0.47}$ | $82.52_{\pm 0.92}$ | $78.05_{\pm 0.93}$ | 12.6 |
| **Heterophily-Designated GNNs** | | | | | | |
| CPGNN | $63.96_{\pm 0.62}$ | $39.79_{\pm 0.77}$ | $52.03_{\pm 5.46}$ | $73.36_{\pm 1.01}$ | $65.96_{\pm 1.95}$ | 40.0 |
| FAGCN | $65.22_{\pm 0.56}$ | $44.12_{\pm 0.30}$ | $88.17_{\pm 0.73}$ | $77.75_{\pm 1.05}$ | $77.24_{\pm 1.26}$ | 31.0 |
| FSGNN | $79.92_{\pm 0.56}$ | $52.74_{\pm 0.83}$ | $90.08_{\pm 0.70}$ | $82.76_{\pm 0.61}$ | $78.86_{\pm 0.92}$ | 18.2 |
| GBK-GNN | $74.57_{\pm 0.47}$ | $45.98_{\pm 0.71}$ | $90.85_{\pm 0.58}$ | $81.01_{\pm 0.67}$ | $74.47_{\pm 0.86}$ | 28.0 |
| GloGNN | $59.63_{\pm 0.69}$ | $36.89_{\pm 0.14}$ | $51.08_{\pm 1.23}$ | $73.39_{\pm 1.17}$ | $65.74_{\pm 1.19}$ | 41.0 |
| GPR-GNN | $64.85_{\pm 0.27}$ | $44.88_{\pm 0.34}$ | $86.24_{\pm 0.61}$ | $72.94_{\pm 0.97}$ | $55.48_{\pm 0.91}$ | 38.4 |
| H2GCN | $60.11_{\pm 0.52}$ | $36.47_{\pm 0.23}$ | $89.71_{\pm 0.31}$ | $73.35_{\pm 1.01}$ | $63.59_{\pm 1.46}$ | 39.6 |
| JacobiConv | $71.14_{\pm 0.42}$ | $43.55_{\pm 0.48}$ | $89.66_{\pm 0.40}$ | $68.66_{\pm 0.65}$ | $73.88_{\pm 1.16}$ | 36.2 |
| **Graph SSMs** | | | | | | |
| GMN | $87.69_{\pm 0.50}$ | $54.07_{\pm 0.31}$ | $91.01_{\pm 0.23}$ | $84.52_{\pm 0.21}$ | – | 11.0* |
| GPS + Mamba | $83.10_{\pm 0.28}$ | $45.13_{\pm 0.97}$ | $89.93_{\pm 0.54}$ | $83.70_{\pm 1.05}$ | – | 25.5* |
| **Ours** | | | | | | |
| MP-SSM | $90.91_{\pm 0.48}$ | $53.65_{\pm 0.71}$ | $95.33_{\pm 0.72}$ | $85.26_{\pm 0.93}$ | $78.18_{\pm 1.34}$ | 2.4 |

- Graph-agnostic temporal predictors, i.e., Historical Average (AV), SVR [99], and FC-LSTM [102], and VAR [71];

- Spatio-temporal GNNs, i.e., DCRNN [68], GConvGRU [92], Graph WaveNet [117], AST-GCN [47], STSGCN [100], GMAN [125], MTGNN [116], AGCRN [7], T-GCN [124], DyGrAE [103], EGCN-O [83], A3T-GCN [6], MPNN LSTM [82], GTS [93], STEP [94],

GC-LSTM [17], DynGESN [76], HMM4G [35], STAEformer [70], RGDAN [36], AdpST-GCN [121], and STD-MAE [39].

## O.2 Datasets statistics

In our experiments, we compute the performance of our MP-SSM on widely used benchmarks for both static and temporal graphs. Specifically, we consider:

- long-range propagation tasks, i.e., the three graph property prediction tasks proposed by [43] ("Diameter", "SSSP", and "Eccentricity") and the "Peptide-func" and "Peptide-struct" tasks from the long-range graph benchmark [32];

- heterophilic tasks, i.e., "Roman-empire", "Amazon-ratings", "Minesweeper", "Tolokers", and "Questions" [86];

- temporal tasks, i.e., "Metr-LA" and "PeMS-Bay" for traffic forecasting [68], and the "Chickenpox Hungary", "PedalMe London", and "Wikipedia math" forecasting tasks introduced by [89].

In Table 11, we report the statistics of the employed datasets.

Table 11: Dataset statistics

|  | Task | Nodes | Edges | Graphs (or Timesteps) | Frequency |
|---|---|---|---|---|---|
| Static | Diameter | 25 - 35 | 22 - 553 | 7,040 | – |
|  | SSSP | 25 - 35 | 22 - 553 | 7,040 | – |
|  | Eccentricity | 25 - 35 | 22 - 553 | 7,040 | – |
|  | Peptide-func | 150.94 (avg) | 307.30 (avg) | 15,535 | – |
|  | Peptide-struct | 150.94 (avg) | 307.30 (avg) | 15,535 | – |
|  | Roman-empire | 22,662 | 32,927 | 1 | – |
|  | Amazon-ratings | 24,492 | 93,050 | 1 | – |
|  | Minesweeper | 10,000 | 39,402 | 1 | – |
|  | Tolokers | 11,758 | 519,000 | 1 | – |
|  | Questions | 48,921 | 153,540 | 1 | – |
| Temporal | Metr-LA | 207 | 1,515 | 34,272 | 5 mins |
|  | PeMS-Bay | 325 | 2,369 | 52,116 | 5 mins |
|  | Chickenpox Hungary | 20 | 102 | 512 | Weekly |
|  | PedalMe London | 15 | 225 | 15 | Weekly |
|  | Wikipedia math | 731 | 27,079 | 1,068 | Daily |

## O.3 Hyperparameter space

In Table 12, we report the grid of hyperparameters employed in our experiments by our method on all the considered benchmarks.

Table 12: The grid of hyperparameters employed during model selection for the graph property prediction tasks (*GPP*), Long Range Graph Benchmark (*LRGB*), heterophilic benchmarks (*Hetero*), and spatio-temporal benchmarks (*Temporal*).

| Hyperparameters | Values | | | |
|---|---|---|---|---|
|  | *GPP* | *LRGB* | *Hetero* | *Temporal* |
| Optimizer | Adam | AdamW | AdamW | AdamW |
| Learning rate | 0.003 | 0.001, 0.0005, 0.0001 | 0.001, 0.0005 ,0.0001 | 0.005, 0.001, 0.0005 ,0.0001 |
| Weight decay | $10^{-6}$ | 0, 0.0001, 0.001 | 0, 0.0001, 0.001 | 0, 0.0001, 0.001 |
| Dropout | 0 | 0, 0.5 | 0, 0.4, 0.5, 0.6, | 0, 0.5 |
| N. recurrences | 1, 5, 10, 20 | 1, 2, 4, 8, 16 | 1, 2, 4, 8, 16 | 1, 2, 4, 8, 16 |
| Embedding dim | 10, 20, 30 | 32,64,128,256 | 32,64,128,256 | 32,64,128,256 |
| N. Blocks | 1, 2 | 1, 2, 4, 8, 16 | 1, 2, 4, 8, 16 | 1, 2, 4, 8, 16 |
| Structure of $\mathbf{U}$ | | $\mathbf{U} = [\mathbf{U}_1, \dots, \mathbf{U}_1]$ | | $\mathbf{U} = [\mathbf{U}_1, \mathbf{U}_2, \dots]$ |

