# OpenReview forum: "Message-Passing State-Space Models: Improving Graph Learning with Modern Sequence Modeling"
_NeurIPS.cc/2025/Workshop/UniReps — UniReps2025_

### Official Review · Reviewer_XEUC · 2025-09-04
**Promising combination of SSMs and MPNNs to get the best of both worlds**

**Confidence:** 3

**Review:**

**Evaluation**

The authors propose a new architecture that combines state space models (SSMs) and message passing neural networks (MPNNs) to address the long-range dependency bottleneck in current MPNNs. The presented experiments show promising results on common benchmarks.

The paper is clear in motivation and the integration idea is well explained. However, there are a few presentation and experimental aspects that could be improved to strengthen the work and make the claims more convincing — especially regarding the connection between the long-range dependency problem and the achieved results.

---

**Pros**
- The general idea of integrating SSMs into MPNNs is well described and motivated.
- The combination of two widely used architectures is promising and has the potential to bring complementary strengths.
- Experiments show competitive results on standard benchmarks.

---

**Cons**
- The paper’s readability would improve significantly with a short formalization of the type of problem the architecture addresses, presented before the MP-SSM description.
- The Method section seems to miss the definition for the matrix \( A \).
- In Table 2, reporting standard deviations would help demonstrate the stability of results with respect to different random seeds.
- While the architecture is motivated by addressing the long-range dependency problem, the current presentation does not clearly support this claim with experimental evidence. If one of the benchmarks specifically challenges existing methods in this regard, it would be valuable to highlight this to substantiate the claim that the design decisions are effective.

---

**Overall Assessment**
This is a promising combination of two common architectures to mitigate the long-range dependency bottleneck. The contribution is relevant and has potential significance. The presentation of results could be improved to more directly connect the initial motivation to the experimental evidence.

**Score:**

4

**Topic Fit:**

1

---

### Official Review · Reviewer_XnNT · 2025-09-15
**Message-Passing State-Space Models: Improving Graph Learning with Modern Sequence Modeling: review**

**Confidence:** 4

**Review:**

The present long abstract proposes message-passing state-space models, a framework in which message-passing graph neural networks (MP-GNNs) are integrated with seq2seq blocks typical of state-space models (SSMs). The authors provide valuable insights on the sensitivity analysis, as the framework allows for an exact characterization of gradient flow via Jacobians, which also shed light on over-squashing and gradient vanishing phenomena. Experiments show a comparison of this model with traditional message-passing and transformers applied to graphs.
The long abstract is well written and the content is theoretically sound; nevertheless, I would like to raise some questions for the authors:
- this is not the first work on graph-learning SSMs, see e.g. [1]. The authors should provide an overview of existing works related to their model and highlight the differences/improvements:
- what is A at line 59? maybe it's implicitly defined as adjacency matrix, but it would be better to explicitly define it;
- it is not described the initialization of X_0;





[1]Li, J., Wu, R., Jin, X., Ma, B., Chen, L., & Zheng, Z. (2024). State space models on temporal graphs: A first-principles study. Advances in Neural Information Processing Systems, 37, 127030-127058.

**Score:**

3

**Topic Fit:**

2

---

### Official Review · Reviewer_hBNJ · 2025-09-15
**Clear and simple work unifying GNNS and SSMs, with strong evaluation, and minor reproducibility and runtime issues.**

**Confidence:** 3

**Review:**

**Summary** The paper presents MP-SSM, a unified model combining state-space dynamics with message passing to improve long-range propagation and achieve strong results on diverse graph learning tasks.


**Strengths**
* Simple and effective approach. Clear formulation unifying graph neural networks and state-space models.
* Claims are well supported both theoretically and empirically. Contributions are clearly documented in the appendix.
* Through evaluation across multiple datasets with fair and extensive comparisons to compatible methods.
* Includes careful examination of known graph-learning limitations (e.g. oversquashing), with empirical evidence of improvement.

**Weaknesses/Questions**
* Sharing code in the final submission would greatly help reproducibility.
* For the runtime results, you might consider using a single, consistent setup, noting how dataset choice affects timings, and clarifying the small mismatch between Fig. 3 and Table 6.

**Score:**

5

**Topic Fit:**

2